# Mechanosensory trichome cells evoke a mechanical stimuli–induced immune response in *Arabidopsis thaliana*

Mamoru Matsumura [1,11], Mika Nomoto [1,2,11✉], Tomotaka Itaya[1,2], Yuri Aratani[3], Mizuki Iwamoto[4], Takakazu Matsuura[5], Yuki Hayashi[1], Tsuyoshi Mori[1], Michael J. Skelly [6], Yoshiharu Y. Yamamoto [7], Toshinori Kinoshita [1,8], Izumi C. Mori [5], Takamasa Suzuki [9], Shigeyuki Betsuyaku [4,10], Steven H. Spoel [6], Masatsugu Toyota [3] & Yasuomi Tada [1,2✉]

Perception of pathogen-derived ligands by corresponding host receptors is a pivotal strategy in eukaryotic innate immunity. In plants, this is complemented by circadian anticipation of infection timing, promoting basal resistance even in the absence of pathogen threat. Here, we report that trichomes, hair-like structures on the epidermis, directly sense external mechanical forces, including raindrops, to anticipate pathogen infections in *Arabidopsis thaliana*. Exposure of leaf surfaces to mechanical stimuli initiates the concentric propagation of intercellular calcium waves away from trichomes to induce defence-related genes. Propagating calcium waves enable effective immunity against pathogenic microbes through the CALMODULIN-BINDING TRANSCRIPTION ACTIVATOR 3 (CAMTA3) and mitogen-activated protein kinases. We propose an early layer of plant immunity in which trichomes function as mechanosensory cells that detect potential risks.

[1] Division of Biological Science, Graduate School of Science, Nagoya University, Aichi, Japan. [2] Center for Gene Research, Nagoya University, Aichi, Japan. [3] Department of Biochemistry and Molecular Biology, Saitama University, Saitama, Japan. [4] Graduate School of Life and Environmental Sciences, University of Tsukuba, Ibaraki, Japan. [5] Institute of Plant Science and Resources (IPSR), Okayama University, Okayama, Japan. [6] Institute of Molecular Plant Sciences, School of Biological Sciences, University of Edinburgh, Edinburgh, UK. [7] Faculty of Applied Biological Sciences, Gifu University, Gifu, Japan. [8] Institute of Transformative Bio-Molecules (WPI-ITbM), Nagoya University, Aichi, Japan. [9] Department of Biological Chemistry, College of Bioscience and Biotechnology, Chubu University, Aichi, Japan. [10] Faculty of Life and Environmental Sciences, Microbiology Research Center for Sustainability, University of Tsukuba, Ibaraki, Japan. [11] These authors contributed equally: Mamoru Matsumura, Mika Nomoto. ✉email: nomoto@gene.nagoya-u.ac.jp; ytada@gene.nagoya-u.ac.jp

nnate immunity is an evolutionarily conserved front line of defense across the plant and animal kingdoms. In plants, pattern-recognition receptors (PRRs), such as leucine-rich repeat receptor-like kinases (LRR-RLKs) and LRR receptor proteins (LRR-RPs), specifically recognize microbe-associated molecular patterns (MAMPs) as non-self molecules, leading to the activation of pattern-triggered immunity (PTI) to limit pathogen proliferation[1,2]. While adapted pathogens have evolved virulence effectors that can circumvent PTI, plants also deploy disease resistance (R) genes, primarily encoding nucleotide-binding LRR proteins, which mount effector-triggered immunity (ETI)[3–5]. ETI often culminates in a hypersensitive response as well as acute and localized cell death at the site of infection accompanied by profound transcriptional changes of defense-related genes to retard pathogen growth[4,5]. These ligand–receptor systems are largely dependent on a transient increase in intracellular calcium concentration ($[Ca^{2+}]_i$), followed by the initiation of phosphorylation-dependent signaling cascades, including mitogen-activated protein kinases (MAPKs) and calcium-dependent protein kinases, that orchestrate a complex transcriptional network and the activity of immune mediators[6,7].

In addition to PTI and ETI, plant immunity can be induced periodically in the absence of pathogen threat, a process controlled by the circadian clock and driven by daily oscillations in humidity as well as light–dark cycles[8–10]. Such responses enable plants to prepare for the potential increased risk of infection at the time when microbes are anticipated to be most infectious. The anticipation of potentially pathogenic microorganisms through sensing climatological changes and their specific detection thus constitute two distinct layers of the plant immune system.

Among the climatological factors that affect the outcome of plant–microbe interactions, rain is a major cause of devastating plant diseases, as fungal spores and bacteria are spread through rain-dispersed aerosols or ballistic particles splashed from neighboring infected plants. Natural raindrops contain bacteria at a concentration of $1.06 \times 10^4$ (/$cm^3$)[11], including plant pathogens such as Pseudomonas syringae[12], Xanthomonas campestris, and Pantoea ananatis[13]. Likewise, raindrops contain fungi such as Alternaria sp., Fusarium sp., Cladosporium sp., Phoma sp., Rhizopus sp., and Botrytis cinerea[14]. In addition, raindrops negatively regulate stomatal closure, which facilitates pathogen entry into leaf tissues[13,15,16]. High humidity, which is usually associated with rain, enhances the effects of bacterial pathogen effectors, such as HopM1, and establishes an aqueous apoplast for aggressive host colonization[17]. These findings suggest that it would be beneficial for plants to recognize rain as an early risk factor for infectious diseases.

How do plants respond to rain? Rain-simulating water spray induces the expression of mechanosensitive TOUCH (TCH) genes as bending stimulation, which induces plant growth retardation[18]. Mechanostimulation affects a variety of plant physiological processes mediated by phytohormones such as auxin, ethylene, and gibberellin[19–22]. Arabidopsis thaliana seedlings exposed to rain-simulating water spray accumulate the immune phytohormone jasmonic acid (JA) to promote the expression of JA-responsive genes[23]. Thus, rain modulates both mechanotransduction and hormone-signaling pathways that could affect the growth and development of plants as well as environmental responses. However, the regulatory mechanisms underpinning the rain-activated signaling pathway have not been fully elucidated.

Here, we report an early layer of the plant immune system evoked by sensing mechanostimulation on the leaf surface: trichomes, hair-like cells, function as mechanosensory cells that mount an effective immune response against both biotrophic and necrotrophic pathogens. When trichomes are mechanically stimulated, intercellular calcium waves are concentrically propagated away from them, MAPKs are activated, and the calcium- and calmodulin-binding

transcription activator (CAMTA)-regulated immune response is initiated. We propose that plants directly recognize rain as a risk factor and evoke a rapid immune response that substantially contributes to early detection of, and protection from, potential pathogens.

## Results and discussion

**Rain and mechanical stimuli induce mechanosensitive genes involved in plant immunity.** To investigate the effect of rain on transcriptional changes in Arabidopsis leaves, we performed transcriptome deep sequencing (RNA-seq) of wild-type Columbia (Col-0) Arabidopsis treated with artificial raindrops (Supplementary Fig. 1a; Methods). After applying only 10 falling droplets, we detected the marked induction of 1050 genes 15 min after treatment [3 biological replicates, $\log_2$ fold changes ($\log_2$FC) $\geq 1$, likelihood ratio test; $P < 0.05$] (Supplementary Data 1). Gene Ontology (GO) analysis of these genes revealed a striking enrichment in categories associated with plant immunity, as evidenced by the expression of major immune regulators, including WRKY DNA-binding protein (WRKY) genes, calmodulin-binding protein 60-like g (CBP60g), MYB domain protein (MYB) genes, ethylene response factor (ERF) genes, and MAP kinase (MPK) genes[24,25] (Fig. 1a and Supplementary Data 1, 2). The touch-induced genes TCH2 and TCH4 were also highly upregulated in response to one falling raindrop (falling) compared to a water droplet placed directly on the leaf surface (static) (Supplementary Fig. 1b). These results suggested that mechanosensation is involved in altering transcriptional activity.

To validate this hypothesis, we mechanically stimulated rosette leaves by gently brushing them 1–10 times along the main veins with a small paintbrush (Supplementary Fig. 1c; Methods) and analyzed the expression profile of the immune regulator WRKY33, which was responsive to raindrops. WRKY33 expression was maximally induced 15 min after brushing the leaves one to four times (Supplementary Fig. 1d). Next, we compared gene expression patterns between leaves that were brushed once and those that received 10 falling raindrops. Both raindrops and brushing strongly upregulated TCH2, TCH4, WRKY33, WRKY40, WRKY53, CBP60g, MYB51, ERF1, and jasmonate-ZIM-domain protein 1 (JAZ1) expression[18,24,25] (Fig. 1b, c), suggesting that raindrops are likely recognized as a mechanical stimulus. This idea was supported by the result that natural rainfall-induced the expression of TCH4 and WRKY33 in Col-0 leaves (Supplementary Fig. 1e).

To comprehensively identify mechanosensitive genes, we performed RNA-seq analysis of leaves brushed once. We identified 1241 genes that were significantly induced 15 min after this treatment relative to control plants (3 biological replicates, $\log_2$FC $\geq 1$, likelihood ratio test; $P < 0.05$) (Supplementary Data 3). These mechanical stimuli (MS)-induced genes were primarily categorized as plant immune responses, such as response to chitin, defense response, and immune system response (Fig. 1d and Supplementary Data 4). We found that 87.3% of raindrop-induced genes and 73.9% of MS-induced genes overlapped (Fig. 1e): this set of 917 genes expressed upon both treatments were enriched for GO categories associated with stress responses (Supplementary Fig. 2). Furthermore, the expression levels of these 917 genes, including major immune regulators, were positively correlated between the two treatments (Pearson correlation coefficient $r = 0.917$) (Fig. 1f). In addition, we observed clear correlations between our profiles and previously published datasets obtained by water spray[23], bending[26], brushing[27], and cotton swabbing treatments[28] (Supplementary Fig. 3 and Supplementary Data 11). Strong correlations observed in Fig. 1f are consistent with the fact that 10 falling droplets

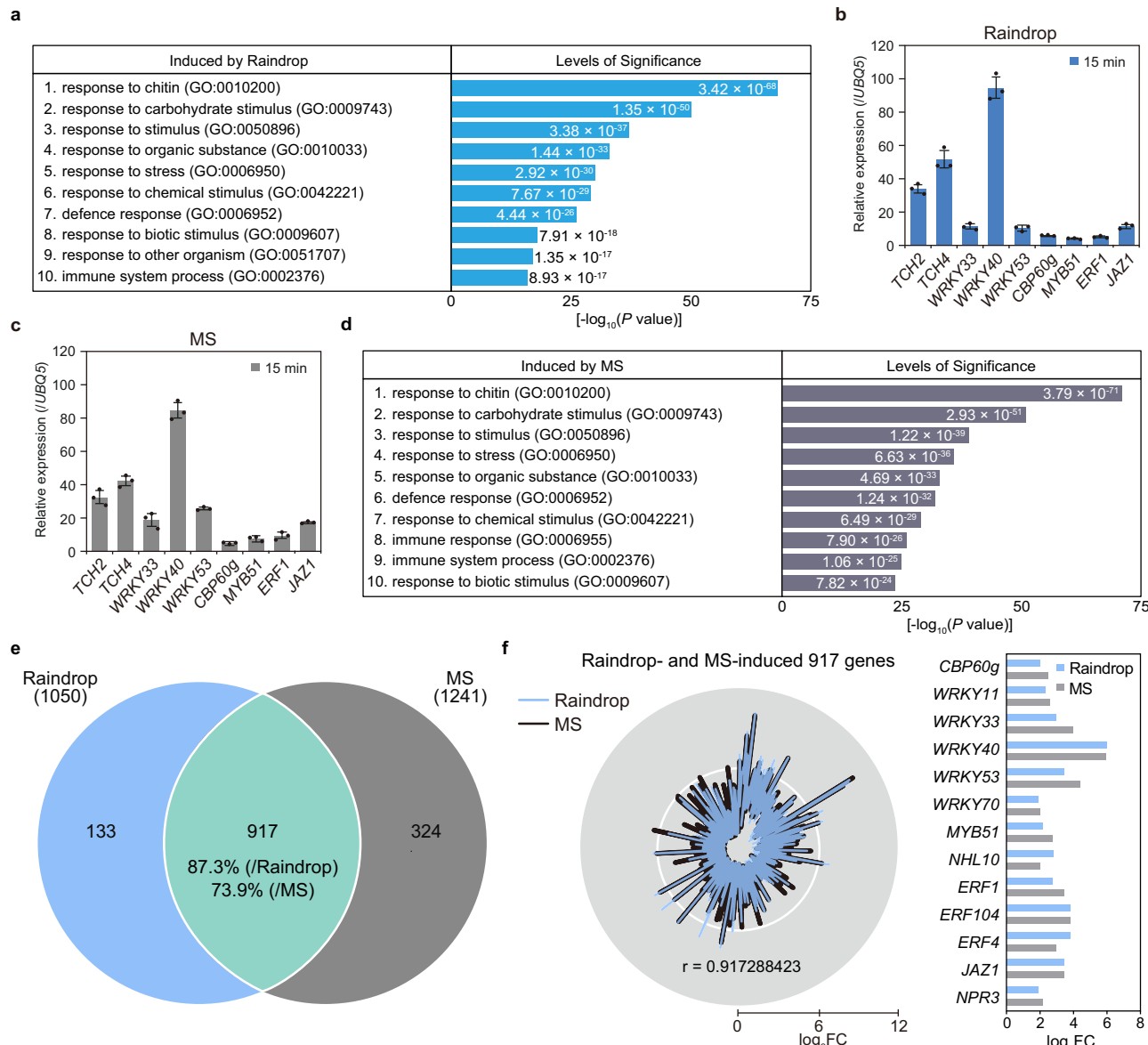

**Fig. 1 Raindrop- and mechanical stimuli (MS)-induced transcriptome profiles highly correlate with each other. a** Enriched Gene Ontology (GO) categories of 1050 raindrop (10 falling droplets)-induced genes in the wild type (Col-0) 15 min after treatment. The top 10 categories are shown in ascending order of $P$ values (one-sided hypergeometric test). **b**, **c** Transcript levels of MS-induced and defense-related genes in 4-week-old Col-0 plants 15 min after being treated with 10 falling droplets (raindrop, **b**) or 1 brushing (MS, **c**), determined by RT-qPCR and normalized to *UBIQUITIN 5* (*UBQ5*). Data are presented as mean ± SD. $n = 6$ plants examined over three independent experiments. Each dot indicates a technical replicate. **d** Enriched GO categories of 1241 MS (1 brushing)-induced genes in Col-0 15 min after treatment. The top ten categories are shown in ascending order of $P$ values (one-sided hypergeometric test). **e** Venn diagram of the overlap between transcriptome datasets from raindrop- and MS-induced genes (likelihood ratio test; $P < 0.05$). **f** Radar chart of intensity compared with mock ($\log_2 FC$) and Pearson correlation coefficient ($r = 0.917288423$) of 917 raindrop- and MS-induced genes (left). Intensities of major immune regulator genes induced by raindrops and MS in RNA-seq analysis ($\log_2 FC$) (likelihood ratio test; $P < 0.05$) (right).

applied the same level of the force intensity as brushing the leaf surface once (Supplementary Fig. 4). Taken together, these transcriptome analyses indicated that falling raindrops stimulate the expression of mechanosensitive genes involved in environmental stress responses, including plant immunity.

**Rain and MS rapidly activate plant immune responses.** To further characterize raindrop-induced genes, we conducted a comparative analysis with published transcriptome datasets. Many raindrop- and MS-induced genes were also expressed

during major plant immune responses, such as those triggered by the immune phytohormones salicylic acid (SA), which is effective against biotrophic pathogens (21%; 193/917 genes), and JA, which mounts immune responses to necrotrophic pathogens (11.8%; 108/917 genes); the bacterial-derived peptide flg22, which activates PTI (37%; 339/917 genes); and the bacterial pathogen *P. syringae* pathovar *maculicola* ES4326 (*Psm* ES4326) (25.8%; 237/917 genes)[1,2,29–33] (Fig. 2a, b). In total, 58.6% (537/917 genes) of raindrop- and MS-induced genes overlapped with those induced in response to different immune elicitors, suggesting that raindrops activate mechanosensitive immune responses.

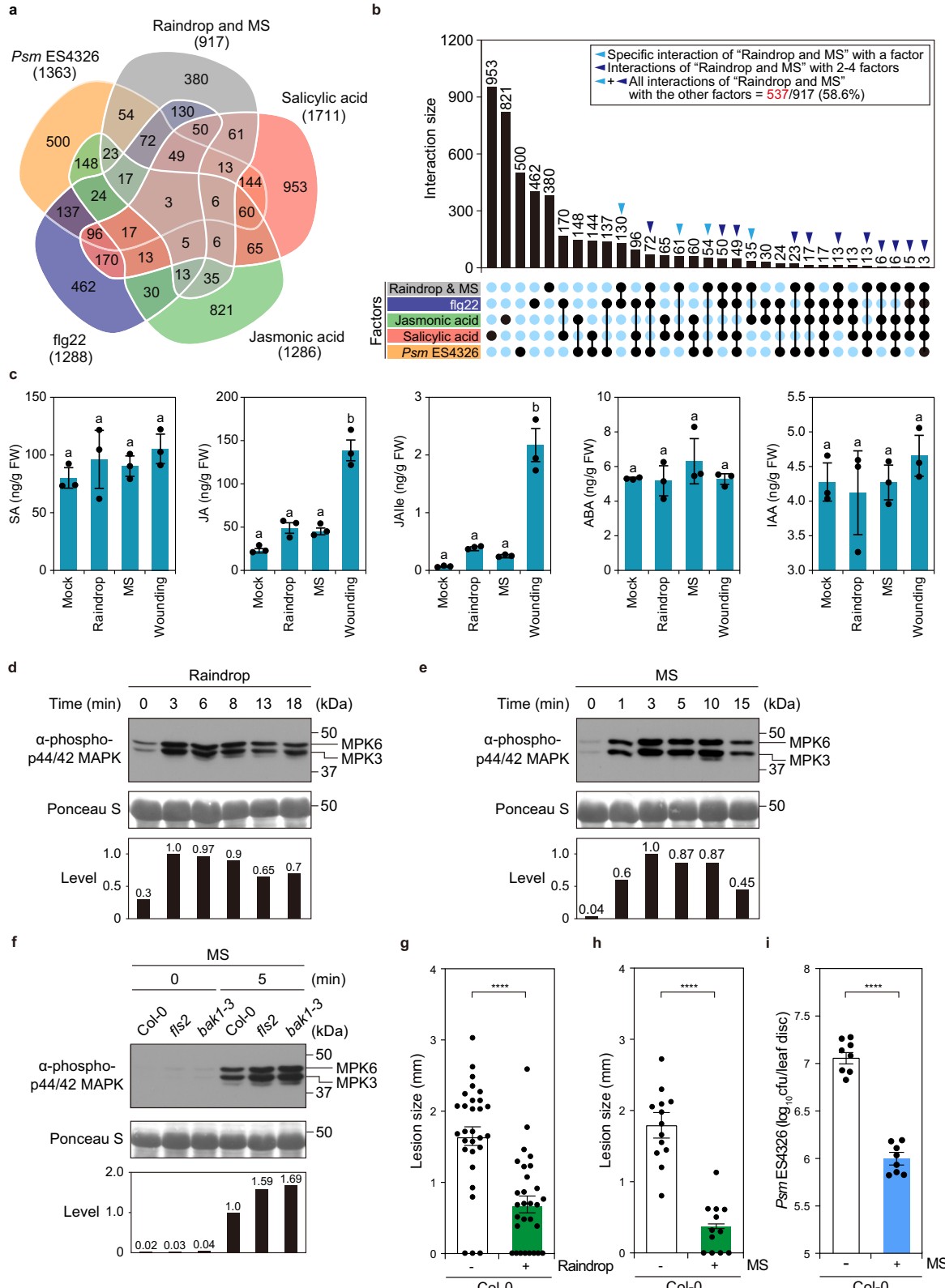

Since stress-responsive gene expression is either positively or negatively regulated by phytohormones, we determined the changes in the accumulation levels of six phytohormones [SA, JA, JA-isoleucine (JA-Ile), abscisic acid (ABA), gibberellic acid 4 (GA₄), and indole-3-acetic acid (IAA)] in leaves treated with 10 falling droplets and in those brushed once. No significant changes in the levels of the phytohormones were observed 5 min and 15 min after treatment (Fig. 2c and Supplementary Fig. 5). However, the slight increase in JA and JA-Ile could explain the observation that 11.8% of raindrop- and MS-induced genes are JA-responsive (Fig. 2a) as previously confirmed by Van Moerkercke et al.[23]. Although 21% of raindrop- and MS-induced

**Fig. 2 Raindrop- and MS-induced mechanosensation triggers defense responses. a, b** Venn diagram (**a**) and the upset plot (**b**) between 917 raindrop- and MS-induced genes and transcriptome datasets obtained from salicylic acid (SA), jasmonic acid (JA), and flg22 (PAMP) treatment and *Pseudomonas syringae* pv. *maculicola* ES4326 infection ($P < 0.05$). Overlap with raindrop- and MS-induced genes: SA, 21%, 193/917 genes; JA, 11.8%, 108/917 genes; flg22, 37%, 339/917 genes; *Psm* ES4326, 25.8%, 237/917 genes; any of the four factors, 58.6%, 537/917 genes. **c** Accumulation of plant hormones (ng/g fresh weight) SA, JA, JA-isoleucine (JA-Ile), abscisic acid (ABA), and indole-3-acetic acid (IAA) 5 min after treatment with 10 falling droplets (raindrop), 1 brushing (MS), or cutting (wounding). Data are presented as mean ± SD. $n = 6$ plants examined over three independent experiments. Each dot indicates a biological replicate. Different letters above bars indicate significant differences (one-sided Tukey's multiple comparison test; $P < 0.05$). **d, e** Raindrop (4 droplets)- (**d**) and MS (4 brushings)-induced (**e**). MAPK activation in Col-0. Total proteins were extracted from 4-week-old plants treated with raindrops and detected by immunoblot analysis with anti-phospho-p44/42 MAPK antibodies. Relative phosphorylation levels are shown below each blot. Similar results were obtained in four independent experiments. **f** MS-induced MAPK activation in Col-0, *fls2*, and *bak1–3*. Total proteins were extracted from 4-week-old plants after 5 min of MS treatment (1 brushing) and detected by immunoblot analysis with anti-phospho-p44/42 MAPK antibodies. Relative phosphorylation levels are shown below each blot. Similar results were obtained in three independent experiments. **g, h** Disease progression of *Alternaria brassicicola* in Col-0 leaves 3 days after inoculation with (+) or without (−) raindrop (10 falling droplets) pretreatment (**g**) or with (+) or without (−) MS (4 brushings) pretreatment (**h**). Error bars represent SE. Asterisks indicate significant difference (two-sided Tukey's *t* test; ****$P < 0.0001$). **i** Growth of *Psm* ES4326 in Col-0 leaves 2 days after inoculation with (+) or without (−) MS (4 brushings) pretreatment. Error bars represent SE. Asterisks indicate significant difference (two-sided Tukey's *t* test; ****$P < 0.0001$). Cfu colony-forming units. $n = 29$ (**g**), (−): $n = 13$, (+): $n = 12$ (**h**), and $n = 8$ (**i**) samples examined over three independent experiments. Each dot indicates a biological replicate.

genes overlap with SA-responsive genes (Fig. 2a), SA levels were not significantly increased in response to raindrops and MS (Fig. 2c). Mechanosensitive genes such as *TCH2*, *TCH4*, *WRKY33*, *WRKY40*, *CBP60g*, *MYB51*, *ERF1*, and *JAZ1*, whose expression was induced 15 min after treatment with MS and raindrops, are presumably regulated independently of phytohormonal responses, as the expression of these genes was not compromised in mutants of the JA-responsive transcription activators, *MYC2*, *MYC3*, and *MYC4*, and the JA-insensitive *jai3-1* (*JASMONATE-INSENSITIVE 3*) (Supplementary Fig. 6). A previous report demonstrated that GA accumulation is reduced by "bending" leaves twice per day for 2 weeks[22]. Here, significant changes in GA levels were not detected upon transient application of raindrops or MS (Supplementary Fig. 5). Therefore, mechanotransduction may branch into phytohormone-dependent and -independent pathways, which are differentially activated by MS depending on their type, intensity, and duration and on which organs perceive the stimulation.

Activation of MAPKs is one of the earliest cellular events and a hallmark of plant immune responses. In particular, PRRs promptly activate a phosphorylation cascade involving MPK3 and MPK6 in response to MAMPs, whereby the downstream immune components of PTI are phosphorylated to promote transcriptional reprogramming[1,6,7,34]. Because 37% of raindrop- and MS-induced genes were also upregulated by flg22 treatment (Fig. 2a), we examined whether a MAPK cascade is activated in responses to raindrops and MS by immunoblot analysis with the anti-p44/42 antibody, which detects phosphorylated MPK3/MPK6[35,36]. A previous study found that MPK4/MPK6 phosphorylation was induced by bending[28,37]. Upon treatment of rosette leaves with four falling raindrops or MS (four brushing), phosphorylation of MPK3/MPK6 was induced within 3 min and remained high for 10 min after each treatment (Fig. 2d, e), indicating that MPK3/MPK6 activation precedes the expression of mechanosensitive genes detected 10 min after MS application (Supplementary Fig. 1d). The kinetics of MS-activated MPK3/MPK6 were reminiscent of those observed upon activation of the PRR protein FLAGELLIN-SENSITIVE 2 (FLS2) and its coreceptor BRI1-associated receptor kinase 1 (BAK1), which are responsible for recognition of the bacterial flg22 epitope[1,34]. Wild-type, *fls2*, and *bak1* mutant plants displayed comparable levels of phosphorylated MPK3/MPK6 in response to MS (Fig. 2f), however, suggesting that FLS2 and BAK1 are not positively involved in raindrop-elicited mechanotransduction.

We then performed a comparative analysis of raindrop- and MS-induced genes against published transcriptome datasets describing the specific and conditional activation of MPK3/MPK6 in transgenic *Arabidopsis* plants carrying a constitutively active variant of tobacco (*Nicotiana tabacum*) *MAP KINASE Cab 2* (*NtMEK2*) under the control of the dexamethasone-inducible promoter[36]. Approximately, 23.9% (328/1374 genes) of both raindrop- and MS-induced genes were upregulated by MPK3/MPK6[36] (Supplementary Fig. 7a and Supplementary Data 5), and these upregulated genes were highly enriched in categories associated with plant immunity (Supplementary Fig. 7b and Supplementary Data 6), suggesting that MAPKs play a critical role in mechanotransduction.

**Rain and MS confer resistance to both biotrophic and necrotrophic pathogens.** We then investigated whether raindrops and MS confer resistance to pathogenic microbes. Raindrops containing the spores of the necrotrophic pathogen *Alternaria brassicicola* Ryo-1 were placed on fully expanded leaves after pretreatment with raindrops or MS for 3 h at an interval of 15 min. Both stimuli significantly suppressed lesion development compared to control plants without pretreatment (Fig. 2g, h). Pretreatment of leaves with MS for 3 h also efficiently protected plants from infection with the biotrophic pathogen *Psm* ES4326 (Fig. 2i). These results confirmed that mechanostimulation induces a PTI-like response to confer a broad spectrum of resistance to both biotrophic and necrotrophic pathogens, as MS activates immune MAPKs and upregulates a large subset of flg22-induced genes. In support of this argument, exposure to the fungal cell wall, chitin, also upregulated 42.1% (386/917 genes) of raindrop-induced genes (Supplementary Fig. 7c).

**Mechanosensitive genes are regulated by calmodulin-binding transcription activator 3.** To dissect rain-induced mechanotransduction, we searched for a conserved *cis*-regulatory element in the promoter sequences of mechanosensitive genes. From an unbiased promoter analysis of the top 300 genes among 917 differentially expressed genes, we obtained the highest enrichment for the CGCG box (CGCGT or CGTGT), which is recognized by CAMTAs that are conserved from plants to mammals[38–42] (Fig. 3a). A similar motif analysis that detects the CAMTA-binding sites among the brushing-induced gene promoters was reported[27]. The *Arabidopsis* transcription factor CAMTA3 [also named signal responsive 1 (SR1)] is a negative regulator of plant immunity; *camta3* null mutants exhibit constitutive expression of defense-related genes and enhanced resistance to virulent *P. syringae* infection[43,44]. CAMTA transcription

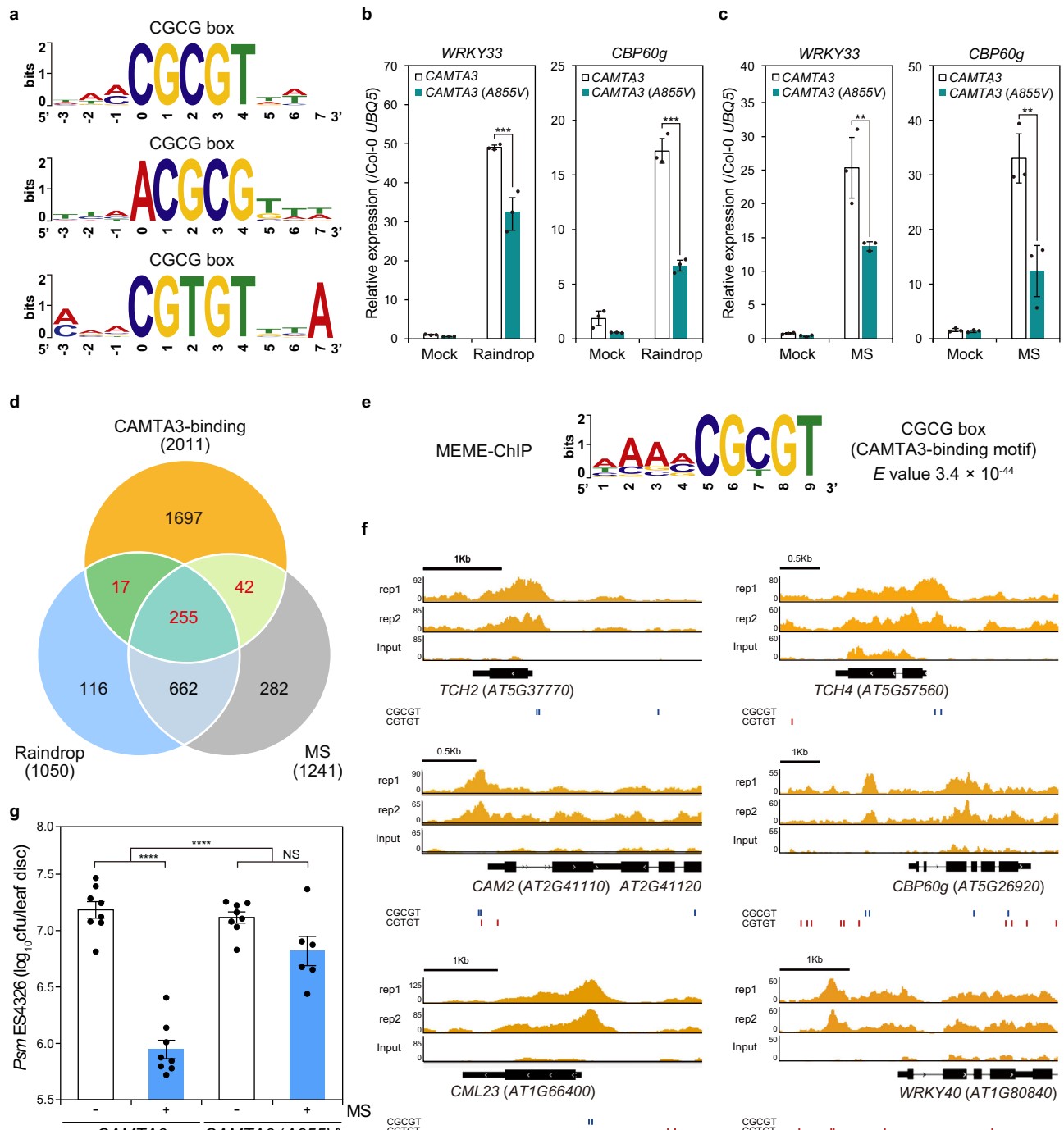

**Fig. 3 MS-induced genes are regulated by CAMTA3. a** Promoter analysis of the top 300 (among 917 genes) raindrop- and MS-induced genes in terms of expression levels revealed that the CAMTA-binding CGCG box [CGC(/T)GT] was overrepresented among these genes. **b**, **c** Transcript levels of *WRKY33* and *CBP60g* in 4-week-old *camta2 camta3 CAMTA3pro:CAMTA3-GFP* (*CAMTA3*) and *camta2 camta3 CAMTA3pro:CAMTA3^A855V^-GFP* [*CAMTA3(A855V)*] plants 15 min after the plants were treated with 1 falling droplet (**b**) or brushed 4 times (**c**), determined by RT-qPCR and normalized to *UBQ5* transcript levels in Col-0 Mock. Data are presented as mean ± SD. Asterisks indicate significant difference (one-sided Tukey's multiple comparison test; **\*\*P* < 0.01, **\*\*\*P* < 0.001). *n* = 6 plants examined over three independent experiments. Each dot indicates a technical replicate. **d** Venn diagram depicting the overlap between genes with CAMTA3-binding sites in their promoters, as determined by ChIP-seq, and raindrop- and MS-induced genes as determined by RNA-seq. A total of 314 genes, shown in red, were identified as CAMTA3-target genes. **e** The CGCG box was identified as an overrepresented motif among the sequence peaks of 314 genes by MEME-ChIP. **f** Localization of CAMTA3 on the promoters of the MS-induced genes *TCH2, TCH4, CAM2, CBP60g, CML23,* and *WRKY40*, as representative of the 314 genes shown in (**d**). Blue and red lines indicate CGCGT and CGTGT, respectively. **g** Growth of *Psm* ES4326 in *camta2 camta3 CAMTA3pro:CAMTA3-GFP* (*CAMTA3*) and *camta2 camta3 CAMTA3pro:CAMTA3^A855V^-GFP* [*CAMTA3(A855V)*] plants 2 days after inoculation with (+) or without (−) MS (4 brushing) pretreatment. Error bars represent SE. Asterisks indicate a significant difference (one-sided Tukey's multiple comparison test and two-way ANOVA; **\*\*\*\*P* < 0.0001). Cfu colony-forming units, NS not significant. *CAMTA3* (−, +), *CAMTA3(A855V)* (−): *n* = 8, *CAMTA3(A855V)* (+): *n* = 6 samples examined over three independent experiments. Each dot indicates a biological replicate.

factors possess a CaM-binding domain and an IQ domain to which CaM binds in a calcium-dependent manner to negate their function (Supplementary Fig. 8a). CAMTA3[A855V] transgenic plants, which possess a mutation in the IQ domain, suppress the constitutive expression of defense-related genes seen in the *camta2 camta3* double mutant and are no longer regulated by calcium-mediated responses[45,46]. In agreement with our promoter analysis, 28.7% of constitutively upregulated genes (309/1075 genes) in the *camta1 camta2 camta3* triple mutant overlapped with raindrop- and MS-induced genes detected in wild-type plants[47] (Supplementary Fig. 8b and Supplementary Data 7). Upon application of raindrops and MS, *WRKY33* and *CBP60g* transcript levels were significantly reduced in plants expressing the *CAMTA3[A855V]* variant compared to a *CAMTA3-GFP* transgenic line expressing a transgene that complemented the phenotype of the *camta2 camta3* mutant (Fig. 3b, c), suggesting that CAMTA3 is involved in mechanotransduction.

To confirm whether CAMTA3 directly targets mechanosensitive genes, we investigated the genome-wide distribution of CAMTA3-binding sites by chromatin immunoprecipitation followed by deep sequencing (ChIP-seq) using *CAMTA3[A855V]-GFP* plants, as the mutant protein stably represses the transcription of CAMTA3-regulated genes. With the aid of model-based analysis of ChIP-seq (MACS2) software, we identified 2641 and 2728 CAMTA3-binding genes, respectively, in two replicates (binomial distribution; $P < 0.05$); about 40% of these peaks are located in the promoter regions and another 30% in gene bodies (Supplementary Fig. 8c and Supplementary Data 8). The overlap between the two replicates highlighted 2011 CAMTA3-targeted genes that included 272 raindrop- and 297 MS-induced genes such as *TCH2*, *TCH4*, and *CBP60g* (Fig. 3d), consistent with our hypothesis that CAMTA3 regulates the transcription of mechanosensitive genes.

To validate the results from the promoter analysis of mechanosensitive genes, we next investigated specific DNA sequences to which CAMTA3 selectively binds by analyzing CAMTA3-binding peaks by Multiple EM for Motif Elicitation (MEME)-ChIP (Methods). We again identified the CGCG box (CGCGT or CGTGT) as the motif with the highest enrichment score ($3.4 \times 10^{-44}$) (Fig. 3e). Subsequent visualization of ChIP-seq profiles via the Integrative Genomics Viewer (IGV)[48] demonstrated that CAMTA3 is primarily enriched at the CGCG boxes of mechanosensitive genes, including *TCH2*, *TCH4*, *CAM2*, *CBP60g*, calmodulin-like 23 (*CML23*), and *WRKY40* (Fig. 3f). GO analysis on 314 CAMTA-targeted genes (Fig. 3d, shown in red) to define the biological functions of these genes showed significant enrichment in categories related to immune and environmental responses (Supplementary Fig. 8d, e). We thus investigated whether CAMTA3 is required for the immune responses. *camta2 camta3 CAMTA3-GFP* transgenic plants effectively mounted an enhanced disease resistance against *P. syringae* in response to MS, while *camta2 camta3 CAMTA3[A855V]-GFP* plants were significantly compromised in resistance (Fig. 3g). These results demonstrate that CAMTA3 negatively regulates plant immune responses by binding to the CGCG box in raindrop- and MS-induced gene promoters and represses the expression of these genes.

Since mechanostimulation rapidly activates MPK3/MPK6 (Fig. 2d, e), we investigated whether CAMTA3 mediates the activation of these MPKs. Using *camta2 camta3 CAMTA3-GFP* and *camta2 camta3 CAMTA3[A855V]-GFP*, we detected the phosphorylation of MPK3/MPK6 independently of CAMTA3 activity (Supplementary Fig. 8f). In addition, the calcium ionophore A23187 clearly induced the phosphorylation of MPK3 and MPK6 (Supplementary Fig. 8g). These results suggested that the mechanotransduction initiated by raindrops and MS may cause a $Ca^{2+}$ influx that negates the repressive effect of CAMTA3 and independently activates the MAPK cascade, as previously proposed[34].

## MS initiates intercellular calcium waves concentrically away from trichomes.

To visualize how mechanostimulation induces the expression of immune genes *in planta*, we generated *Arabidopsis* transgenic lines with the promoter sequences of *WRKY33* and *CBP60g* driving the expression of nucleus-targeted enhanced *yellow fluorescent protein* (*YFP-NLS*) (*WRKY33pro:EYFP-NLS* and *CBP60gpro:EYFP-NLS*). *WRKY33* expression is regulated by both MPK3/MPK6 and CAMTA3, while *CBP60g* is not mediated by MPK3/MPK6 (Supplementary Data 5). When half leaves were gently brushed (Supplementary Fig. 1c), we detected YFP fluorescence in the *WRKY33pro:EYFP-NLS* and *CBP60gpro:EYFP-NLS* transgenic lines as localized, clustered groups of cells only in the brushed half (Fig. 4a and Supplementary Fig. 9a). Closer inspection of the stimulated regions revealed that both genes were induced in cells surrounding trichomes, hair-like structures projecting outward from the epidermal surface (Fig. 4b, c and Supplementary Fig. 9b, c).

Trichomes function as chemical and physical barriers against insect feeding and are likely involved in drought tolerance and protection against ultraviolet irradiation[49,50]. Mechanostimulation of a single trichome induces $Ca^{2+}$ oscillations within the proximal skirt cells that surround the base of trichomes[51], suggesting that the mechanical force could be focused on only skirt cells (Supplementary Fig. 10). However, since mechanostimulation by raindrops and MS confers resistance to pathogens in whole leaves, we hypothesized that trichomes activate a $Ca^{2+}$ signal in a large area of leaves, as shown in Fig. 4a.

To visualize changes in cytosolic $Ca^{2+}$ concentrations ($[Ca^{2+}]_{cyt}$) induced by MS on the leaf surface, we used transgenic *Arabidopsis* expressing the GFP-based $[Ca^{2+}]_{cyt}$ indicator GCaMP3[52,53]. Leaf brushing induced a marked increase of $[Ca^{2+}]_{cyt}$ in the surrounding leaf area of trichomes 1 min after stimulation (Fig. 4d and Supplementary Movie 1). Flicking a single trichome with a silver chloride wire triggered an intercellular calcium wave that propagated concentrically away from the trichome and surrounding skirt cells at a speed of 1.0 μm/s (Fig. 4e, f and Supplementary Movie 2). This pattern showed striking consistency with the area of induction observed with the *WRKY33pro:EYFP-NLS* and *CBP60gpro:EYFP-NLS* reporters (Fig. 4a, b and Supplementary Fig. 9a, b). The base of trichomes exhibited a rapid and transient increase in $[Ca^{2+}]_{cyt}$ before the concentric propagation of calcium waves was initiated (Fig. 4g and Supplementary Movie 3).

## Trichomes are mechanosensory cells activating plant immunity.

To investigate the possible involvement of trichomes in mechanosensation in *Arabidopsis* leaves and activation of the immune response, we observed calcium waves using the knockout mutant of *glabrous 1* (*GL1*)[54], which lacks trichomes. The *gl1* mutant exhibits effective basal resistance comparable to that of wild-type Col-0 plants[54,55], and its local resistance to *Psm* ES4326 and *A. brassicicola* Ryo-1 is similar to that of Col-0 plants (Supplementary Fig. 11). However, the trichome-defective *gl1*, *gl3* (*glabrous 3*), and *ttg1* (*transparent testa glabra 1*) mutants lost cuticle and systemic acquired resistance (SAR)[55], a secondary immune response in uninfected distal tissues after primary infection in a local site, while basal levels of immunity were not dramatically affected[55] (Fig. 5 and Supplementary Fig. 11). The lack of SAR in these trichome mutants is associated with impairment of cuticle formation but not of trichomes. *cer1-1* (*eceriferum 1*) and *cer3-1* (*WAX 2/eceriferum 3*) mutants, which have defects in synthesizing cuticular wax but form normal

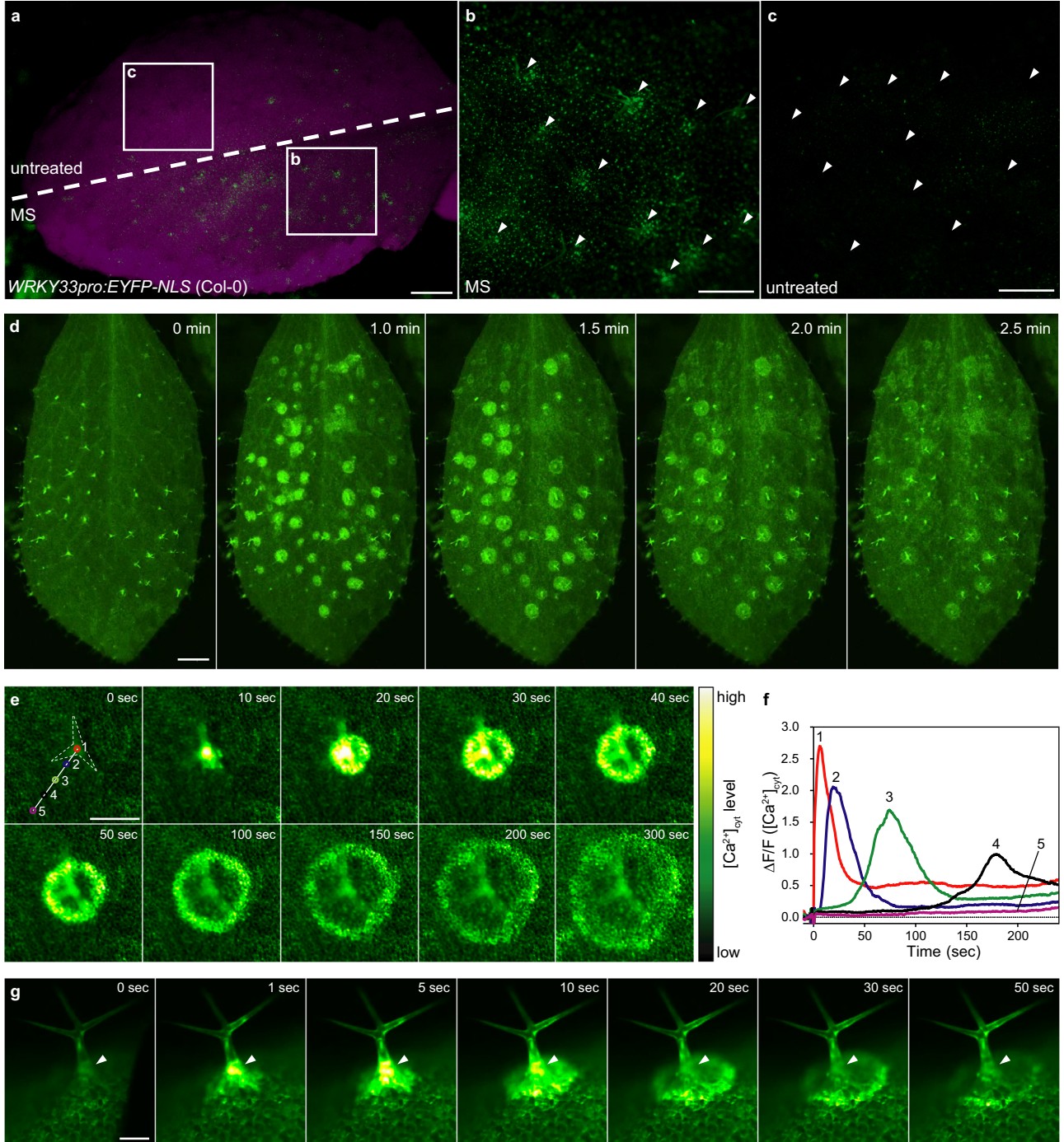

**Fig. 4 Trichomes initiate intercellular calcium waves. a–c** YFP fluorescence in a whole leaf from *WRKY33pro:EYFP-NLS* (Col-0) with (MS, bottom half) or without brushing (untreated, top half) (**a**), along with zoomed-in views of brushed (**b**), and untreated (**c**) areas. Arrowheads indicate trichomes (**b**, **c**). Scale bars, 0.5 mm (**a**), 0.3 mm (**b**, **c**). Similar results were obtained in three independent experiments. **d** Ca$^{2+}$ imaging using *35Spro:GCaMP3* (Col-0). The leaf surface of a 4-week-old plant was treated with MS by brushing. MS-induced intercellular calcium waves propagated concentrically from trichomes. Scale bar, 1.0 mm. See also Supplementary Movie 1. **e** Ca$^{2+}$ imaging using *35Spro:GCaMP3* (Col-0). A single trichome from a 2-week-old seedling was flicked with a silver chloride wire. MS-induced intercellular calcium waves propagated concentrically from the trichome (dashed outline). Scale bar, 0.2 mm. See also Supplementary Movie 2. Similar results were obtained in three independent experiments. **f** [Ca$^{2+}$]$_{cyt}$ changes at sites indicated by numbers in (**e**). Similar results were obtained in three independent experiments. **g** Side view of a trichome whose neck was flicked with a silver chloride wire. MS-induced intercellular Ca$^{2+}$ influx was transiently observed in the trichome base (arrowheads) followed by the formation of circular waves. Scale bar, 0.1 mm. See also Supplementary Movie 3. Similar results were obtained in three independent experiments.

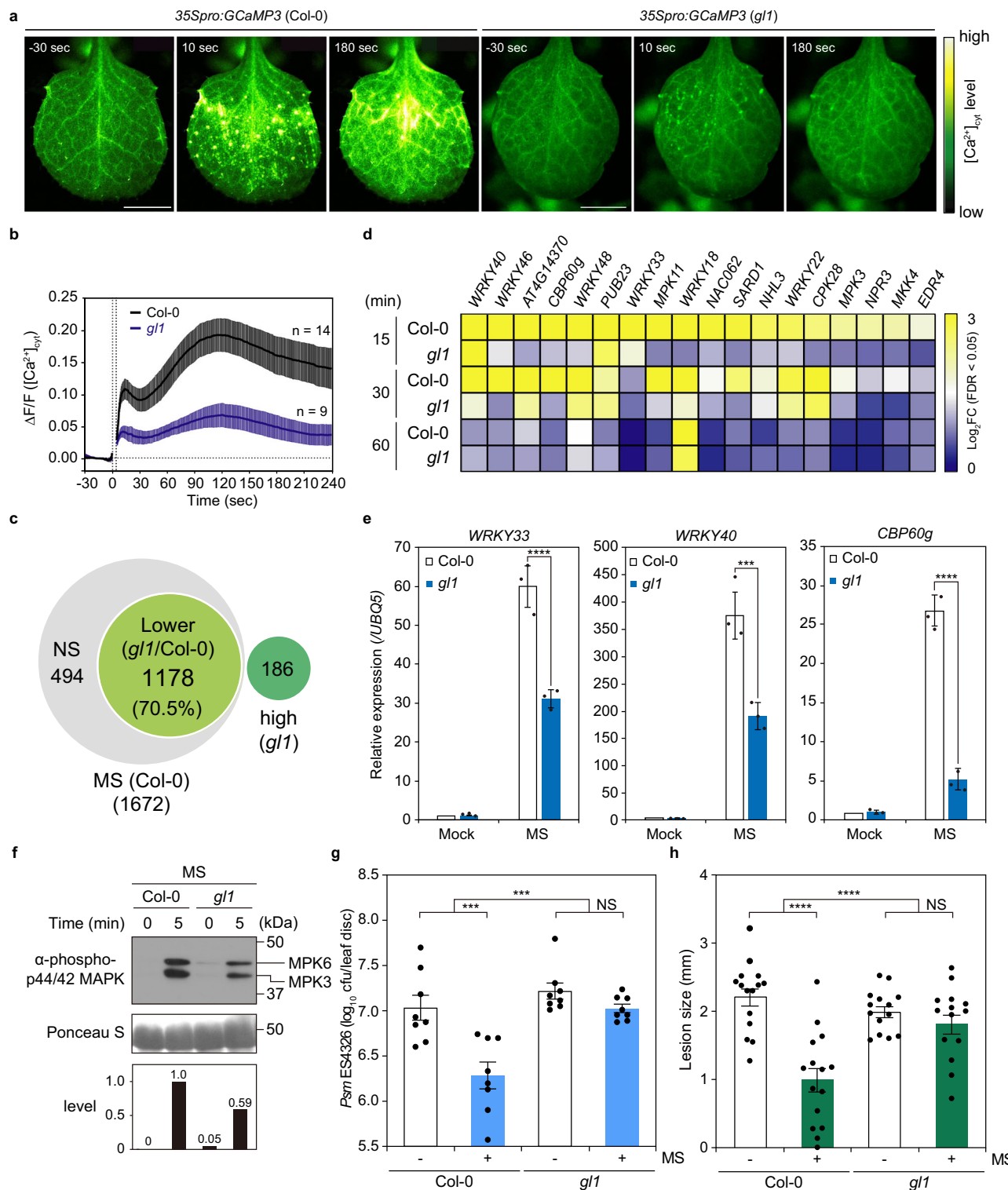

trichomes, fail to induce SAR[56]. Therefore, the immunodeficient phenotype of *gl* and *ttg* mutants is due to disordered cuticle formation. To investigate whether trichome-mediated immunity is compromised in *cer1* and *cer3* mutants as in *gl1*, we performed the pathogen test using *A. brassicicola* Ryo-1 after mechanical stimulation of these mutants (Supplementary Fig. 12). Pretreatment with brushing significantly suppressed lesion development. These data strongly indicate that the lack of trichome, but not cuticle, in *gl* mutants is responsible for failed MS-induced immunity. The mechanostimulation-induced propagation of

concentric calcium waves was compromised in the *gl1* mutant (Fig. 5a, b and Supplementary Fig. 13), confirming that trichomes are true MS sensors and initiate calcium waves (Supplementary Movies 4 and 5). Furthermore, approximately 70.5% of mechanosensitive genes were expressed in a trichome-dependent manner (3 biological replicates, $\log_2\text{FC} \geq 1$, likelihood ratio test; $P < 0.05$) (Fig. 5c, Supplementary Fig. 14a and Supplementary Data 9), and transcript levels of 18 representative mechanosensitive immune genes were markedly lower at all time points in the *gl1* mutant than they were in the wild type in RNA-seq

**Fig. 5 Trichomes are mechanosensory cells. a** $Ca^{2+}$ imaging using *35Spro:GCaMP3* (Col-0) and *35Spro:GCaMP3* (*gl1*). Leaf surfaces were exposed to MS by brushing. MS-induced calcium waves were compromised in the *gl1* mutant. See also Supplementary Movies 4 and 5. Scale bars, 0.5 mm. **b** $[Ca^{2+}]_{cyt}$ signature of (**a**). **c** Venn diagram of transcriptome datasets for MS-induced genes in Col-0 and *gl1* (likelihood ratio test; $P < 0.05$). NS, not significant. Lower, fold change (FC) (*gl1*)/FC (Col-0) < 0.5. High, MS (*gl1*)/Mock (*gl1*), $\log_2 FC \geq 1$ in *gl1* (likelihood ratio test; $P < 0.05$). **d** Heatmap of differentially expressed defense-related genes obtained from transcriptome datasets from Col-0 and *gl1* plants treated with MS (4 brushings). **e** Transcript levels of *WRKY33*, *WRKY40*, and *CBP60g* in 4-week-old Col-0 and *gl1* plants 15 min after treatment with brushing 4 times, determined using RT-qPCR and normalized to *UBQ5*. Data are presented as mean ± SD. Asterisks indicate significant difference (one-sided Tukey's multiple comparison test; ***$P < 0.001$, ****$P < 0.0001$). $n = 6$ plants examined over three independent experiments. Each dot indicates a technical replicate. **f** MS-induced MAPK activation in Col-0 and *gl1*. Total proteins were extracted from 4-week-old leaves 5 min after MS (4 brushings) treatment and detected by immunoblot analysis with anti-phospho-p44/42 MAPK antibodies. Relative phosphorylation levels are shown below each blot. Similar results were obtained in three independent experiments. **g** Growth of *Psm* ES4326 in Col-0 and *gl1* leaves 2 days after inoculation with (+) or without (−) MS (4 brushings) pretreatment. Error bars represent SE. Asterisks indicate a significant difference (one-sided Tukey's *t* test and two-way ANOVA; ***$P < 0.001$). Cfu colony-forming units, NS not significant. $n = 8$ samples examined over three independent experiments. Each dot indicates a biological replicate. **h** Disease progression of *Alternaria brassicicola* in Col-0 and *gl1* leaves 3 days after inoculation with (+) or without (−) MS (4 brushings) pretreatment. Error bars represent SE. Asterisks indicate a significant difference (one-sided Tukey's *t* test and two-way ANOVA; ****$P < 0.0001$). NS not significant. Col-0: $n = 15$, *gl1*: $n = 14$ samples examined over three independent experiments. Each dot indicates a biological replicate.

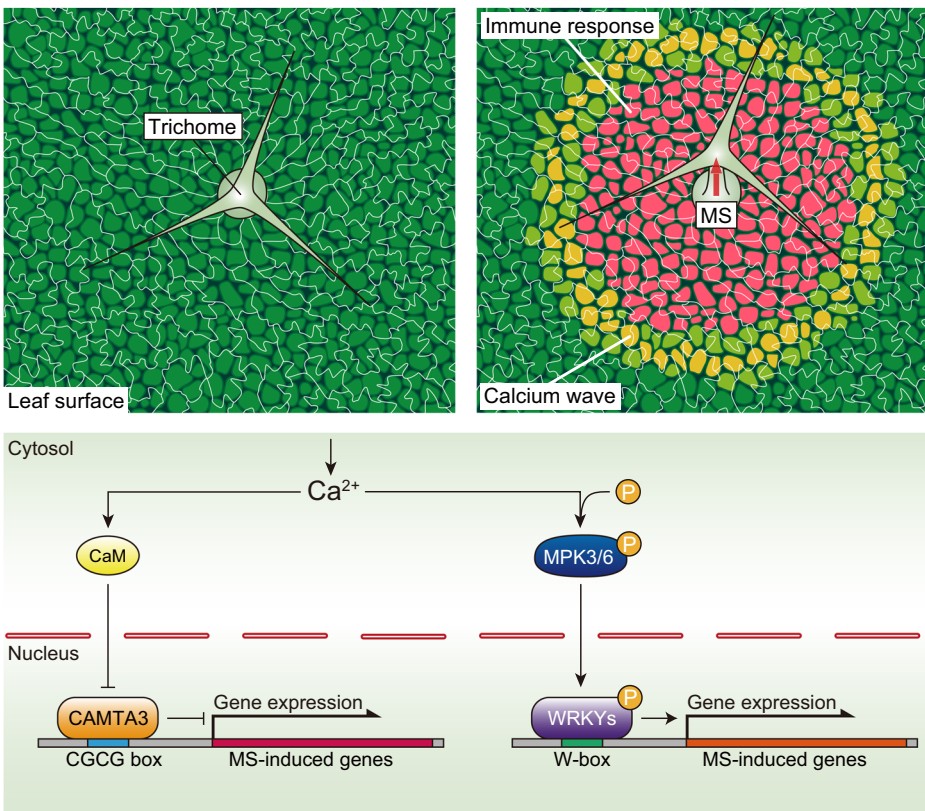

**Fig. 6 Mechanosensory trichomes evoke an immune response.** Model showing how trichomes directly sense the mechanical impact of raindrops as an emergency signal in anticipation of possible infections. Mechanosensory trichome cells initiate intercellular calcium waves in response to MS. $[Ca^{2+}]_{cyt}$ initiates the de-repression of $Ca^{2+}$/CaM-dependent CAMTA3 and activates the phosphorylation of MPK3 and MPK6, thereby inducing WRKY-dependent transcription.

analysis of leaves brushed four times (Fig. 5d), suggesting that trichomes serve as a sensor of this type of MS on the leaf surface. Similarly, compared to wild-type plants, the transcription of MS- and raindrop-induced *WRKY33*, *WRKY40*, and *CBP60g*, as well as the activation of MPK3/MPK6 by MS, was also significantly reduced in the *gl1* mutant (Fig. 5e, f and Supplementary Fig. 14b). Moreover, MS-induced resistance against *Psm* ES4326 infection was abrogated in the *gl1* mutant (Fig. 5g). As with *Psm* ES4326, the application of MS to wild-type plants prior to inoculation with *A. brassicicola* significantly limited lesion development, whereas the *gl1* mutant did not show enhanced disease resistance in response to MS (Fig. 5h). Mechanically stimulating leaves by

bending them back and forth 10 times induces JA-dependent resistance to *B. cinerea* in both Col-0 and *gl1* mutant[57], suggesting that bending directly stimulates cells inside the leaf.

Our work highlights an early layer of plant immunity that is triggered by an unexpected function of trichomes as mechanosensory cells. When trichomes are mechanically stimulated, intercellular calcium waves are concentrically propagated away from the trichomes, followed by the activation of CAMTA3-dependent immune responses (Fig. 6). Rapid phosphorylation of MAPKs also is a prerequisite for mechanosensitive gene expression[28], as MPK3/MPK6 mediate the phosphorylation of mechanosensitive WRKY33 for its activation[58,59]. This notion is

supported by the finding that the expression of 252 of 917 raindrop- and MS-induced genes are mediated by MPK3/MPK6 (Supplementary Fig. 7a), and their promoter sequences possess the W-box (TTGACC) for WRKYs as the most enriched cis-regulatory elements (Supplementary Fig. 15). The molecular mechanism by which calcium mediates the activation of MPK3/MPK6 has yet to be elucidated.

Mechanostimulation by repeatedly bending leaves confers resistance to the necrotrophic pathogen *B. cinerea* via JA accumulation[57]. In addition, a subset of JA-responsive genes upregulated by water spray is mediated by MYC2/MYC3/MYC4 transcription factors[23]. These observations strongly indicate that mechanosensation causes profound JA-dependent changes in gene transcription, promoting plant immune responses to necrotrophic pathogens. The JA- and MYC-dependent pathway does not play a major role in the expression of mechanosensitive *TCH* genes, however, indicating that mechanotransduction is regulated by other signaling pathways. In addition, we investigated the requirement for MS-induced immunity in the defense mutants *fls2*, *bak1-3*, *npr1-3* (nonexpresser of PR genes 1), and *myc234*, and found that FLS2 and NPR1 could be involved in the signaling pathway (Supplementary Fig. 16). Our work demonstrated that raindrops and MS only partially activate the JA signal but rather strongly induce a PTI-like response via the $Ca^{2+}$- and CAMTA3-dependent pathway, which is highly effective against both necrotrophs and biotrophs (Fig. 2g–i). Because rain disseminates diverse pathogens with different parasitic strategies, including fungi, bacteria, and viruses[60,61], it is reasonable that plants perceive raindrops as a risk factor and activate broad-spectrum resistance.

Plants possess mechanosensory cells with a variety of functions, such as flower antennas of *Catasetum* species for pollination, tentacles of *Drosera rotundifolia* for insect trapping, root hairs of *Arabidopsis* for water tracking, and red cells of *Mimosa pudica* for evading herbivores[62]. The carnivorous Venus flytrap (*Dionaea muscipula*) captures insects by sensing mechanostimulation via sensory hairs on leaf lobes[63]. To monitor diverse MS applied to plants, several sensing mechanisms have been proposed that include detection of cell wall components, distortion of the plasma membrane, and displacement of the plasma membrane against the cell wall[64]. In all these systems, a transient increase in $[Ca^{2+}]_{cyt}$ is thought to play a pivotal role in short- and long-term responses. Two successive stimulations of sensory hairs of the flytrap are required to meet the threshold of $[Ca^{2+}]_{cyt}$ for rapid closure of the leaf blade[63,65]. As the trichome on the leaf surface is widely found in many land plants, there may be a common intercellular network of cell–cell communication that initiates calcium waves for activating immune responses.

## Methods

**Plants**. *A. thaliana* accession Columbia-0 (Col-0) was the background for all plants used in this study. *WRKY33pro:EYFP-NLS* (Col-0) and *CBP60gpro:EYFP-NLS* (Col-0) were generated as described in the subsection "Promoter-reporter imaging"[66]. *35Spro:GCaMP3* (Col-0)[53], *camta2 camta3 CAMTA3pro:CAMTA3-GFP*, and *camta2 camta3 CAMTA3pro:CAMTA3^{A855V}-GFP* were provided from Michael F. Thomashow (Michigan State University)[45,47]. *gl1* [Col(gl1)] was obtained from Lehle Seeds (TX, USA) and was previously designated as *gl-1* mutant. The *Arabidopsis* mutants *fls2* (SALK_093905) and *bak1-3* (SALK_034523) were obtained from the Arabidopsis Biological Resource Center (ABRC). *35Spro:GCaMP3* was introduced into the *gl1* mutant background by crossing. The selection of homozygous lines was performed by genotyping using primers listed in Supplementary Data 10. Plants were grown on soil (peat moss; Super Mix A and vermiculite mixed 1:1) at 22 °C under diurnal conditions (16-h-light/8-h-dark cycles) with 50–70% relative humidity. *WRKY33pro:EYFP-NLS* (Col-0) and *CBP60gpro:EYFP-NLS* (Col-0) were sown on soil and grown in a growth room at 23 °C in constant light[66]. *35Spro:GCaMP3* (Col-0) and *35Spro:GCaMP3* (*gl1*) were grown on Murashige and Skoog (MS) plates [1× MS salts, 1% (w/v) sucrose, 0.01% (w/v) myoinositol, 0.05% (w/v) MES, and 0.5% (w/v) gellan gum, pH 5.8][53,67].

**Artificial raindrop treatment**. Reverse osmosis water was kept in a 500 mL beaker until the water temperature reached room temperature (22 °C). A transfusion set (NIPRO Infusion Set TI-U250P, Nipro, Osaka, Japan) was installed on a steel stand with the beaker at a height of 1.2 m (H-type Stand I3, As One, Osaka, Japan) and was adjusted to release 13 μL water droplets (Supplementary Fig. 1a). In this setting, the applied mechanical energy to the leaf surface is equivalent to one in which 5.8 μL of raindrops reach a terminal velocity of 6.96 m/s[68]. This size raindrop is frequently observed in nature; thus, the impact of simulated rain is comparable with that of true rain[68]. The adaxial side of leaves from 4-week-old plants was treated with 10 droplets for RNA-seq and 1, 4, or 10 droplets for quantitative RT-PCR (RT-qPCR). The adaxial side of leaves from 4-week-old plants was treated with 1 falling or static droplet (Supplementary Fig. 1b). Sample leaves were collected 15 min after treatment and stored at −80 °C until use.

**Brush treatment**. The adaxial side of leaves from 4-week-old plants was brushed once or 4 times for RNA-seq and RT-qPCR along the main veins at an angle of 30–40° (KOWA nero nylon drawing pen flat 12, Kowa, Aichi, Japan) (Supplementary Fig. 1c). Sample leaves were collected 15, 30, and 60 min after treatment for RNA-seq and 15 min after treatment for RT-qPCR and stored at −80 °C until use.

**RNA-seq library construction**. Total RNA was extracted from 80 to 100 mg frozen samples using Sepasol-RNA I Super G (Nacalai Tesque, Kyoto, Japan) and the TURBO DNase free kit (Thermo Fisher Scientific, IL, USA) according to the manufacturer's protocols. Total RNA was further purified with the RNeasy RNA Isolation Kit (QIAGEN, Hilden, Germany) and assessed for quality and quantity with a Nanodrop 2000 spectrophotometer (Thermo Fisher Scientific). We used 1 μg total RNA for mRNA purification with NEBNext Oligo d(T)$_{25}$ (NEBNext poly(A) mRNA Magnetic Isolation Module; New England Biolabs, MA, USA), followed by first-strand cDNA synthesis with the NEBNext Ultra II RNA Library Prep Kit for Illumina (New England Biolabs) and NEBNext Multiplex Oligo for Illumina (New England Biolabs) according to the manufacturer's protocols. For the analysis of raindrop- and MS-induced gene expression, the amount of cDNA was determined on an Agilent 4150 TapeStation System (Agilent, CA, USA). cDNA libraries were sequenced as single-end reads for 81 nucleotides on an Illumina NextSeq 550 (Illumina, CA, USA). The reads were mapped to the *Arabidopsis thaliana* reference genome (TAIR10, http://www.arabidopsis.org/) on the web (BaseSpace, Illumina, https://basespace.illumina.com/). Pairwise comparisons between samples were performed with the EdgeR[69] package on the web (Degust, https://degust.erc.monash.edu/).

For the comparative analysis of differentially expressed genes between leaves in the *gl1* mutant and Col-0, the amount of cDNA was determined by the QuantiFluor dsDNA System (Promega, WI, USA). cDNA libraries were sequenced as single-end reads for 36 nucleotides on an Illumina NextSeq 500 (Illumina). The reads were mapped to the *A. thaliana* reference genome (TAIR10) via Bowtie2[70] with the options "--all --best --strata". Pairwise comparisons between samples were performed with the EdgeR package in the R program[69]. Enrichment of GO categories for biological processes was determined using BiNGO (http://www.psb.ugent.be/cbd/papers/BiNGO/Home.html) (one-sided hypergeometric test; $P < 0.05$)[71]. Pearson correlation coefficient of the expression levels between two transcriptome profiles was calculated with Excel function (PEARSON function) ($r = 0$–$0.2$: no correlation, $r = 0.2$–$0.4$: weak correlation, $r = 0.4$–$0.7$: slightly correlated, $r = 0.7$–$1.0$: correlated).

**Quantification of the force density**. The abaxial side of leaves from 4-week-old plants was physically attached to the measuring pan of the electronic balance QUINTIX224-1S (Sartorius Lab Instruments GmbH & Co., Göttingen, Germany) with surgical tape (3 M Company, MN, USA). The adaxial side of the leaf was treated with 1 falling droplet or brushed once (shown in the above subsections). The peak weight applied to the leaf surface was obtained as the force. The force per unit area (N/m²) is converted from the peak weight (kg) and the contact area of the brush tip ($5.6 \times 10^{-5}$ m²) or raindrop ($9.73 \times 10^{-6}$ m²).

**Re-analysis of immune-related transcriptome datasets**. We used the following public transcriptome datasets for comparative analysis with the RNA-seq data obtained in this study: 10-day-old *Arabidopsis* seedlings treated with 1 μM flg22 for 30 min (Array Express; E-NASC-76)[29], 8-day-old *Arabidopsis* seedlings treated with 40 μM chitin for 1 h (Gene Expression Omnibus; GSE74955), leaves from 4-week-old *Arabidopsis* plants inoculated with *Pseudomonas syringae* pv. *maculicola* (*Psm*) ES4326 (24 h post inoculation) (GSE18978), 2-week-old *Arabidopsis* seedlings treated with 0.5 mM SA or 50 μM JA for 24 h (DNA Data Bank of Japan; DRA003119)[33], 2-week-old *Arabidopsis* seedlings grown with MS medium and treated with water spray for 10 min or 25 min (E-MTAB-8021)[23], leaves from 4-week-old *Arabidopsis* plants treated with bending back and forth manually for 30 min (accession is now not available)[26], leaves from 4-week-old *Arabidopsis* plants treated with brushing for 30 min (NCBI BioProject; PRJNA473032)[27], 2-week-old *Arabidopsis* seedlings grown with MS medium and treated with cotton swabbing for 30 min (accession is now not available)[28], 12-day-old *Arabidopsis*

*GVG-NtMEK2DD* seedlings treated with 2 μM DEX for 0 and 6 h (NCBI Sequence Read Archive; SRP111959)[36], and 4-week-old *Arabidopsis camta1 camta2 camta3* triple mutant (GSE43818)[47] (Supplementary Data 11). The overlaps between differentially expressed genes in each transcriptome dataset were evaluated as Venn diagrams (http://bioinformatics.psb.ugent.be/webtools/Venn/) and Upset plot (https://asntech.shinyapps.io/intervene/).

**RT-qPCR.** Total RNA was extracted from 30–40 mg leaf tissues with Sepasol-RNA I Super G and the TURBO DNase free kit (Thermo Fisher Scientific) according to the manufacturer's protocols, followed by reverse transcription with the Prime-Script RT reagent kit (Takara Bio, Shiga, Japan) using oligo dT primers. RT-qPCR was performed on the first-strand cDNAs diluted 20-fold in water using KAPA SYBR FAST qPCR Master Mix (2×) kit (Roche, Basel, Switzerland) and gene-specific primers in a LightCycler 96 (Roche). Primer sequences are listed in Supplementary Data 10.

**Quantification of plant hormones.** The adaxial side of leaves from 4-week-old plants was treated with 10 raindrops (raindrop), brushed once (MS), or cut (wounding). Sample leaves (0.07–0.1 g) were collected 5 min or 15 min after treatment and stored at −80 °C until use. Frozen samples were ground to a fine powder, mixed with 4 mL extraction solution [80% (v/v) acetonitrile, 1% (v/v) acetic acid], and stored for 1 h at 4 °C to extract plant hormones. After centrifugation at 3000 g for 10 min, the supernatants were evaporated in a vacuum centrifugal evaporator EC-57CS (Sakuma, Tokyo, Japan) and dissolved in 1% (v/v) acetic acid. Samples were loaded onto a reverse-phase solid-phase extraction cartridge (Oasis HLB 1 cc; Waters Corporation, MA, USA). The cartridge was washed with 1 mL 1% acetic acid, and hormones were eluted with 2 mL extraction solution. The eluent was evaporated to obtain samples in 1 mL 1% acetic acid and subjected to cation exchange chromatography on an Oasis MCX 1 cc extraction cartridge (Waters Corporation). The acidic fraction was eluted with 1 mL extraction solution. The acidic eluent was analyzed for SA, and the remaining fraction was evaporated, dissolved in 1% acetic acid, and loaded onto an Oasis WAX 1-cc extraction cartridge (Waters Corporation). The cartridge was washed with 1% acetic acid and the remaining hormones were eluted with extraction solution. The elute was analyzed for JA, JA-Ile, ABA, IAA, and GA$_4$. The contents of these hormones were quantified using liquid chromatography–electrospray tandem mass spectrometry (LC–ESI–MS/MS) (triple quadrupole mass spectrometer with 1260 high-performance LC, G6410B; Agilent Technologies Inc., CA, USA) equipped with a ZORBAX Eclipse XDB-C18 column (Agilent Technologies Inc.)[72].

**ChIP assay.** Approximately 0.7 g of 2-week-old *camta2 camta3 CAMTA3pro:-CAMTA3A855V-GFP* seedlings was fixed in 25 mL 1% formaldehyde under vacuum for three cycles of 2 min ON/2 min OFF using an aspirator (SIBATA, Tokyo, Japan). Subsequently, 1.5 mL of 2 M glycine was added to quench the cross-linking reaction under vacuum for 2 min. The samples were then washed with 50 mL double-distilled water and stored at −80 °C until use. Frozen samples were ground to a fine powder with a mortar and pestle in liquid nitrogen and dissolved in 2.5 mL nuclei extraction buffer (10 mM Tris-HCl pH 8.0, 0.25 M sucrose, 10 mM MgCl$_2$, 40 mM β-mercaptoethanol, protease inhibitor cocktail)[33,73]. Samples were filtered through two layers of Miracloth (Calbiochem, CA, USA) and centrifuged at 17,700 g at 4 °C for 5 min. The pellets were resuspended in 75 μL nuclei lysis buffer [50 mM Tris-HCl pH 8.0, 10 mM EDTA, 1% (w/v) SDS]. After incubation first at room temperature for 20 min and then on ice for 10 min, the samples were mixed with 225 μL ChIP dilution buffer without Triton [16.7 mM Tris-HCl pH 8.0, 167 mM NaCl, 1.2 mM EDTA, 0.01% (w/v) SDS]. Chromatin samples were sonicated for 35 cycles of 30 sec ON/30 s OFF using a Bioruptor UCW-201 (Cosmo Bio, Tokyo, Japan) to produce DNA fragments, followed by the addition of 375 μL ChIP dilution buffer without Triton, 200 μL ChIP dilution buffer with Triton [16.7 mM Tris-HCl pH 8.0, 167 mM NaCl, 1.2 mM EDTA, 0.01% (w/v) SDS, 1.1% (w/v) Triton X-100], and 35 μL 20% (w/v) Triton X-100. After centrifugation at 17,700 g at 4 °C for 5 min, 900 μL solubilized sample was split into two 2.0 mL PROKEEP low-protein-binding tubes (Watson Bio Lab USA, CA, USA) and incubated with 0.75 μL anti-GFP antibody (Cat#ab290; Abcam, Cambridge, UK) (1:600 dilution) or Rabbit IgG-Isotype Control (Input) (Cat#ab37415; Abcam) (1:600 dilution) for 4.5 h with gentle rocking, and an 18 μL aliquot was used as the input control. Then, samples from *camta2 camta3 CAMTA3pro:CAMTA3A855V-GFP* were mixed with 50 μL of a slurry of Protein A agarose beads (Upstate, Darmstadt, Germany) and incubated at 4 °C for 1 h with gentle rocking. Beads were washed twice with 1 mL low-salt wash buffer [20 mM Tris-HCl pH 8.0, 150 mM NaCl, 2 mM EDTA, 0.1% (w/v) SDS, 1% (w/v) Triton X-100], twice with 1 mL high-salt wash buffer [20 mM Tris-HCl pH 8.0, 500 mM NaCl, 2 mM EDTA, 0.1% (w/v) SDS, 1% (w/v) Triton X-100], twice with 1 mL LiCl wash buffer [10 mM Tris-HCl pH 8.0, 0.25 M LiCl, 1 mM EDTA, 1% (w/v) sodium deoxycholate, 1% (w/v) Nonidet P-40], and twice with 1 mL TE buffer [10 mM Tris-HCl pH 8.0, 1 mM EDTA]. After washing, beads were resuspended in 100 μL elution buffer [1% (w/v) SDS, 0.1 M NaHCO$_3$] and incubated at 65 °C for 30 min. For the input controls, 41.1 μL TE buffer, 8.7 μL 10% (w/v) SDS, and 21 μL elution buffer were added to 18 μL of each solubilized sample. Both supernatant and input samples were mixed with 4 μL of 5 M NaCl and incubated at 65 °C overnight to reverse the cross-

linking, followed by digestion with 1 μL Proteinase K (20 mg/ml) (Invitrogen, CA, USA) at 37 °C for 1 h. ChIP samples were mixed with 500 μL Buffer NTB and purified using the PCR clean-up gel extraction kit following the manufacturer's instructions (MACHEREY-NAGEL, Düren, Germany).

**ChIP-seq library construction.** ChIP-seq libraries for the input and two biological replicates were constructed from 2 ng purified DNA samples with the NEB Ultra II DNA Library Prep Kit for Illumina (New England Biolabs) according to the manufacturer's instructions. The amount of DNA was determined on an Agilent 4150 TapeStation System (Agilent). All ChIP-seq libraries were sequenced as 81-nucleotide single-end reads using an Illumina NextSeq 550 system.

**Analysis of ChIP-seq.** Reads were mapped to the *Arabidopsis thaliana* reference genome (TAIR10, http://www.arabidopsis.org/) using Bowtie2 with default parameters[70]. The Sequence Alignment/Map (SAM) file generated by Bowtie2 was converted to a Binary Alignment/Map (BAM) format file by SAMtools[74]. To visualize mapped reads, Tiled Data Files file were generated from each BAM file using the igvtools package in the IGV[48]. ChIP-seq peaks were called by comparing the IP with the Input using MACS2 with the "-p 0.05 -g 1.19e8" option (binomial distribution; $P < 0.05$)[75]. The peaks were annotated using the nearest gene using the Bioconductor and the ChIPpeakAnno packages in the R program, from which we identified 2011 genes detected in both biological replicates. Enrichment of GO categories of the set of 314 genes overlapping between raindrop- and MS-induced genes for biological processes was determined using BiNGO (http://www.psb.ugent.be/cbd/papers/BiNGO/Home.html)[71]. Sequences of the peaks were extracted from the *Arabidopsis thaliana* genome as FASTA files with BEDtools[76]. To identify the candidates of CAMTA3-binding motifs, the FASTA files were subjected to MEME-ChIP with the default parameters (-meme-minw 6-meme-maxw 10)[77], and a density plot of the distribution of the motifs were generated.

**Immunoblot analysis for detection of MPK3 and MPK6 phosphorylation.** The adaxial side of leaves from 4-week-old plants was brushed four times or treated with four raindrops, and samples (0.1–0.15 g) were snap-frozen in liquid nitrogen. Total proteins were extracted in protein extraction buffer [50 mM Tris-HCl pH 7.5, 150 mM NaCl, 2 mM DTT, 2.5 mM NaF, 1.5 mM Na$_3$VO$_4$, 0.5% (w/v) Nonidet P-40, 50 mM β-glycerophosphate, and proteinase inhibitor cocktail] and centrifuged once at 6000 g, 4 °C, for 20 min and twice at 17,000 g, 4 °C for 10 min. The supernatant was mixed with SDS sample buffer [50 mM Tris-HCl pH 6.8, 2% (w/v) SDS, 5% (w/v) glycerol, 0.02% (w/v) bromophenol blue, and 200 mM DTT] and heated at 70 °C for 20 min. The protein samples were subjected to SDS-PAGE electrophoresis and transferred onto a nitrocellulose membrane (GE Healthcare, IL, USA). The membrane was incubated with an anti-phospho-p44/42 MAPK polyclonal antibody (Cat#9101; Cell Signaling Technology, MA, USA) (1:1000 dilution) and goat anti-rabbit IgG(H + L)-HRP secondary antibody (Cat#170-6515; BIO-RAD, CA, USA) (1:2000 dilution). The bands for MPK3/6 were visualized using chemiluminescence solution mixed 5:1 with ImmunoStar Zeta (FUJIFILM Wako Chemicals, Osaka, Japan) and SuperSignal West Dura Extended Duration Substrate (Thermo Fisher Scientific). The Rubisco bands were stained with Ponceau S (Merck Sharp & Dohme Corp., NJ, USA) as a loading control. The phosphorylation levels of MPK3 and MPK6 were quantified with the blot analysis plug-in in ImageJ (https://imagej.nih.gov/ij/).

**Treatment with the calcium ionophore A23187.** Twelve-day-old Col-0 seedlings were treated with 50 μM calcium ionophore A23187 (Sigma-Aldrich Co., MO, USA) for 15, 30, and 60 min. Samples were processed for the phosphorylation of MPK3 and MPK6 as described in the "Immunoblot analysis for detection of MPK3 and MPK6 phosphorylation" section. The leaf tissue was stored at −80 °C until use.

**Promoter–reporter imaging.** The 3.0-kbp promoters for *WRKY33* and *CBP60g*, both of which covered the previously analyzed respective regulatory sequences, were amplified from Col-0 genomic DNA by PCR and cloned into the pENTR/D-TOPO vector (Invitrogen). The promoter regions were recombined using Gateway technology into the binary vector pBGYN. The resulting pBGYN-pWRKY33-EYFP-NLS and pBGYN-pCBP60g-YFP-NLS vectors were introduced into *A. tumefaciens* GV3101 (pMP90) and then into *Arabidopsis* Col-0 plants using the floral dip method. A representative homozygous line was selected for each construct for further detailed analyses.

Promoter-reporter imaging was performed using an MA205FA automated stereomicroscope (Leica Microsystems, Wetzlar, Germany) and DFC365FX CCD camera (Leica Microsystems) in 12-bit mode. Chlorophyll autofluorescence and YFP fluorescence were detected through Texas Red (TXR) (excitation 560/40 nm, extinction 610 nm) and YFP (excitation 510/20 nm, extinction 560/40 nm) filters (Leica Microsystems). To image fluorescence emanating from the *WRKY33pro:EYFP-NLS* (Col-0) and *CBP60gpro:EYFP-NLS* (Col-0) plants[66], the leaves of 3-week-old *Arabidopsis* plants were brushed 10 times at an interval of 15 min for 2 h or left untreated.

**Promoter analysis**. The statistical analysis for overrepresented transcriptional regulatory elements across transcriptome datasets described above was calculated using a prediction program[41]. The $P$ values were calculated using Statistical Motif Analysis in Promoter or Upstream Sequences (https://www.arabidopsis.org/tools/bulk/motiffinder/index.jsp). Figures of promoter motif sequences are generated with WebLogo (https://weblogo.berkeley.edu/logo.cgi).

**Real-time [Ca$^{2+}$]$_{cyt}$ imaging**. We used 4-week-old and 3-week-old plants expressing the GFP-based cytosolic Ca$^{2+}$ concentration ([Ca$^{2+}$]$_{cyt}$) indicator GCaMP3[52,53]. To image the fluorescence from the GCaMP3 reporter (in Col-0 and *gl1*) in whole leaves, the adaxial sides of leaves from 4-week-old plants were brushed. To monitor the calcium waves propagating from trichomes, a single trichome from a 2-week-old seedling was flicked with a silver chloride wire. Samples were imaged with a motorized fluorescence stereomicroscope (SMZ-25; Nikon, Tokyo, Japan) equipped with a 1× objective lens (NA = 0.156, P2-SHR PLAN APO; Nikon) and an sCMOS camera (ORCA-Flash 4.0 V2; Hamamatsu Photonics, Shizuoka, Japan)[53].

To detect the accumulation levels of GCaMP3 protein in *35Spro:GCaMP3*/Col-0 and *35Spro:GCaMP3*/*gl1*, leaves from 4-week-old plants (0.1–0.15 g) were snap-frozen in liquid nitrogen. Total proteins were extracted in protein extraction buffer [50 mM Tris-HCl pH 7.5, 150 mM NaCl, 2 mM DTT, 0.5% (w/v) Nonidet P-40, and proteinase inhibitor cocktail] and centrifuged once at 6000 g, 4 °C, for 20 min and twice at 17,000 g, 4 °C for 10 min. The supernatant was mixed with SDS sample buffer [50 mM Tris-HCl pH 6.8, 2% (w/v) SDS, 5% (w/v) glycerol, 0.02% (w/v) bromophenol blue, and 200 mM DTT] and heated at 70 °C for 20 min. The protein samples were subjected to SDS-PAGE electrophoresis and transferred onto a nitrocellulose membrane (GE Healthcare). The membrane was incubated with an anti-GFP antibody (Cat#ab290; Abcam) (1:4000 dilution) and a goat anti-rabbit IgG(H + L)-HRP secondary antibody (Cat#170-6515; BIO-RAD) (1:2000 dilution). GCaMP3 proteins were visualized using chemiluminescence solution mixed 5:1 with ImmunoStar Zeta (FUJIFILM Wako Chemicals) and SuperSignal West Dura Extended Duration Substrate (Thermo Fisher Scientific). Rubisco proteins were stained with Ponceau S (Merck Sharp & Dohme Corp.) as the loading control. The protein levels of GCaMP3 were quantified with the blot analysis plug-in in ImageJ.

**Propidium iodide staining**. A stock solution of 10 mM propidium iodide (PI) was prepared with phosphate-buffered saline pH7.5. Rosette leaves of 4-week-old Col-0 plants were cut into 5 mm squares, floated in a glass petri dish with 20 μM PI solution, and incubated for 1 h at room temperature. Stained tissues were observed under the all-in-one fluorescence microscope (BZ-X800; Keyence Corporation, Osaka, Japan) equipped with a 20× objective lens (CFI S Plan Fluor LWD ADM 20XC, Nikon) and TRITC dichroic mirror (excitation 545/25 nm, extinction 605/70 nm) (KEYENCE).

**Bacterial infection**. MS was applied to the adaxial leaf surface of 4-week-old plants by brushing 4 times at an interval of 15 min for 3 h. Sample leaves were then inoculated by infiltration, using a plastic syringe (Terumo Tuberculin Syringe 1 mL; TERUMO), with *Psm* ES4326 (OD$_{600}$ = 0.001) resuspended in 10 mM MgCl$_2$. Bacterial growth was measured 2 days after inoculation as described previously[78].

**Fungal infection**. *Alternaria brassicicola* strain Ryo-1 was cultured on 3.9% (w/v) potato dextrose agar plates (PDA; Becton, Dickinson and Company, NJ, USA) for 4–20 days at 28 °C in the dark. After incubation of the agar plates for 3–7 days under ultraviolet C light, a conidial suspension of *A. brassicicola* was obtained by mixing with RO water[79]. The adaxial side of leaves from 4-week-old plants was treated with 10 droplets or MS by brushing 4 times at an interval of 15 min for 3 h, followed by spotting with 5 μL conidia suspension ($2 \times 10^5$ per mL) of *A. brassicicola* on the adaxial side of leaves. Inoculated plants were placed at 22 °C under diurnal conditions (16-h-light/8-h-dark cycles) with 100% relative humidity. The lesion size of fungal infection was measured with ImageJ 3 days after inoculation.

**Statistics and reproducibility**. GraphPad Prism 9 (GraphPad Software, CA, USA) was used for all statistical analyses. One-sided or two-sided Tukey's multiple comparison test, one-sided Šídák's multiple comparison test, or two-way analysis of variance (two-way ANOVA) were used for multiple comparisons. In all graphs, asterisks indicate statistical significance tested by one-sided or two-sided Tukey's multiple comparison test, one-sided Šídák's multiple comparison test, or two-way ANOVA (multiple groups).

**Reporting summary**. Further information on research design is available in the Nature Research Reporting Summary linked to this article.

## Date availability
The authors declare that all data supporting the findings of this study are available within this article and its Supplementary Information files. RNA-seq and ChIP-seq data have

been deposited in the DDBJ Sequence Read Archive at the DNA Data Bank (http://www.ddbj.nig.ac.jp/) with the accession numbers DRA011970, DRA009248, and DRA011123. Source data are provided with this paper.

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

## Acknowledgements

We thank M.F. Thomashow for the seeds of CAMTA3 variants, S. Ishiguro for the seeds of cer1-1 and cer3-1, Y. Takano for *Alternaria brassicicola* Ryo-1, N. Nakayama, T. Kawasaki, T. Matsushita, and G. Goshima for discussion of the work. This work was supported by JSPS KAKENHI Grant Numbers JP23120520, JP25120718, and JP18K19334 (to Y.T.), JP15H05955 (to Y.T. and T.K.), and JP19H05363, JP21H00366 (to M.N.) and by Cooperative Research Grant #1707 of the Plant Transgenic Design Initiative (PTraD) by Gene Research Center, Tsukuba-Plant Innovation Research Center, the University of Tsukuba (to Y.T.).

## Author contributions

M.M., M.N., S.H.S., and Y.T. designed the research. M.M. and T.I. established the artificial rain device. M.M. optimized the protocols for artificial raindrop and brush treatment. M.M. and M.N. constructed the Illumina sequencing libraries for RNA-seq and ChIP-seq. T.S. performed RNA-seq and analysis. M.N. performed the ChIP-seq and analysis of CAMTA3. T.M. and I.C.M. performed the quantification of phytohormones. M.M., Y.H., and T.K. performed the detection of MPK3 and MPK6 phosphorylation.

M.M., T.M., and Y.Y.Y. performed the promoter analysis. Y.A. and M.T. generated *35Spro:GCaMP3* (*gl1*) plants, and M.N., M.T., and Y.A. visualized real-time $[Ca^{2+}]_{cyt}$. M.I. and S.B. generated the transgenic lines *WRKY33pro:EYFP-NLS* (Col-0) and *CBP60gpro:EYFP-NLS* (Col-0). M.M., M.I., and S.B. performed promoter-reporter imaging. M.N. and M.M. performed the rest of the experiments. M.M., M.N., M.J.S., S.H.S., and Y.T. wrote the paper with input from all authors.

## Competing interests

The authors declare no competing interests.
