## [Peer Review File · Nature Communications]

Mechanosensory trichome cells evoke a mechanical stimuli-induced immune response in *Arabidopsis thaliana*REVIEWER COMMENTS

Reviewer #1 (Remarks to the Author):

In this manuscript authors report that Arabidopsis trichomes can act as mechanosensory structures that can activate a wide range of immune responses. It is a comprehensive study and connects the mechanical input plants experience in nature (raindrops) to the immune resistance, identifying signalling factors (Ca²⁺, MAPKs, hormones, transcription factors, and their target genes) in between. The writing is concise and clear and successfully summarizes (what must have been) years of work succinctly.

I have few points to suggest changes that can improve the manuscript further.

1) This report emphasizes raindrop as a possible trigger of immune responses through its mechanical stimulating effect. While this is plausible, this begs for further experimentations such as: a) analysis of pathogen content of raindrops, and b) comparison of artificial raindrop experiment and MS treatment to plants under natural rain (more pathogen response?). This additional sample is wanted in all analysis in Fig 1, 2, and 4. Alternatively, it can be written from a slightly shifted angle that mechanical stimulation of the trichome (including raindrops) triggers defence response. This shift would also emphasize the broader applicability of the findings; they have uncovered general mechano-induction of defence responses that is trichome-dependent.

2) Since mechanical stimulation of the trichomes seems to be the critical trigger, it would be helpful to have explicit estimation of how much stress or strain the trichome experiences in receiving a raindrop or the particular types of brushing.

3) Note that the mechano stimulation used in this study is quite specific – seemingly selected with the mechano-sensation by trichomes in mind (raindrops and brushing). There are other types of physiologically relevant mechanical stimulation, such as bending of the leaf or stem, stomping of the whole plants, wind effects, and so on, and this should be accounted in some sentences (e.g., L157: it's not just duration or intensity; L302 – main sensor of this type of MS, or say senses mechanical stimulation on the leaf surface).

4) How much more susceptible are gl mutants to pathogens?

5) L344: My recollection is that trichomes are not monophyletic in all plant species – so conserved is a strong word. Replace with found?

6) More details of the RNA-seq is needed. How many replicates per sample type? Read depth?

Reviewer #2 (Remarks to the Author):

In this work by Matsumara et al., a wide range of experiments and meta-analyses are presented that add further insight into how plants respond to mechanical stimulation. The title focuses on the role of trichome cells in immune responses, but this topic is only addressed in the last two results sections, and various other data is presented earlier on in the manuscript. Though in principle all the findings are of interest, in my opinion many of the findings are not conceptual advances and are rather confirmatory or adding incremental advances. Moreover, the manuscript is written in a style that would suggest the uninformed reader that many findings are novel, while they are to a large extent conceptually part of the published literature. The manuscript is clearly written and the figures are well presented.

Major comments.

1) The manuscript presents a lot of findings that are confirmatory or supplemental to existing studies. For instance, the concept that the impact of water drops on leaves causes an overlapping response to 'classical' mechanical touching e.g. with a brush is well-known already for decades (e.g. Braam et al., Cell, 1990), and confirmed by many studies. The dose response analysis of number of drops to gene expression shows one drop is largely sufficient and two drops results in very similar effects as 10 drops. The first section of the results describes an RNA seq analysis of the response after 15 minutes to impact of 10 drops of water from about 120 cm high onto the leaf. This is in principle a comparable experiment to the RNA-seq analysis performed previously with water spray bottles at various time points from 10-120 minutes (Van Moerkercke et al., 2019, PNAS). Brushing RNA-seqs have also been performed and published recently e.g. Xu et al., Plant Journal, 2019 (<https://pubmed.ncbi.nlm.nih.gov/30537160/>) and Wang et al., PNAS, 2018 (<https://www.ncbi.nlm.nih.gov/pmc/articles/PMC6205429/>). The comparative analysis of the authors own-data with these published data sets would be useful here. I could also not find details regarding fold-change/p-value cut-offs and multiple testing corrections for the authors new data set in the results section or methods.

2) The meta-analysis comparing the water drop/MS transcriptomes with published data sets focusing on e.g. flg22 pseudomonas and SA/JA is interesting, but again not very novel, as similar analyses have been done. Furthermore the methods section lacks a lot of detail on how these experiments were performed

on different platforms were analysed (e.g. software packages, parameters for fold-changes/p-values). A table giving the precise experimental conditions would be needed, as many treatments like spraying with chemicals like SA/Flg22 would trigger mechanoresponses in themselves. Percentage overlaps between data sets are presented, but there is no statistical analysis whether such overlaps are statistically speaking meaningful or possible by random distributions.

3) The hormone analysis again largely confirms the findings of Van Moerkercke et al., 2019 that mainly JA and JA-Ile are early touch responsive. I could however not find indications of statistical significance in the graphs of Fig 1C. The very early 5 min time point is interesting, but further time points should have been taken to match the RNAseq data. Also a strong drop in ABA concentration was measured 10-25 minutes after water spray previously by Van Moerkercke et al., which likely is caused by the water excess due to water spray/drops. This effect was not found here after 5 minutes, but may become apparent some minutes later. As stated by the authors in the introduction, this may affect immune responses e.g. via stomatal opening allowing pathogens to enter, and would be thus important to confirm using the counted water drop system. The GA data should be shown (even if supplementary).

4) I strongly disagree with the statement on line 152-154, saying that most mechanosensitive genes, whose expression is induced after 5 minutes, are presumably regulated independently of phyto hormones. First of all, it is unclear which specific genes induced after 5 minutes are referred to here, as the authors' RNA-seq data set is done on 15 minutes and the selected WRKY33 reporter gene is also not yet consistently induced by rain drops after 5 minutes (it is induced after 10 min)(Supp figure 1d). Furthermore, JA/JA-Ile are clearly induced by water drops/MS at the hormone level even after 5 min (though stats are lacking) in Figure 1c.

5) The data showing that MPK3/6 are phosphorylated by raindrops/MS within minutes is interesting, but was again already shown for MPK4/6 one minute after MS by Ishimura et al., 2008 (<https://onlinelibrary.wiley.com/doi/full/10.1046/j.1365-313x.2000.00913.x>). The finding that phosphorylation does not seem to operate via FLS2 and BAK1 receptors is of interest. Also the suggestion on line 182 that MAPKs play a critical role in mechanotransduction is very circumstantial based on the presented data, as well as previously suggested in many other studies. Wang et al., 2018, PNAS also showed the importance of phosphorylation of e.g. TREP1 for mechanosignalling.

6) Mechanostimulation has extensively been shown to trigger resistance to necrotrophic pathogens and insect pests (e.g. Chehab et al., 2012, *Current Biology*; Choi et al. 2017, *Sci Rep*; summarised by Ghosh et al., 2021, *J Exp Bot*). The addition of a biotrophic pathogen like *Pseudomonas* to this list is of course interesting, but again confirmatory. The similarity with chitin-induced transcriptomic responses was also reported already by Xu et al., 2019, *Plant J*.

7) The finding that CAMTA3 transcription factor is involved in mechanosensing is probably among the most interesting finding of the study. The authors perform meta-analysis of previously reported microarray data and find an overlap between camta123 regulated genes and their own raindrop/MS transcriptomes. An overrepresentation of the camta DNA binding site in touch induced promoters (Fig 3a) was already reported by Xu et al., 2019, *Plant J*. The Chip-seq on CAMTA3 is of clear interest and forms one of the most interesting data sets of this paper. The description of the different mutants/complementations of CAMTA transgenics could be improved to clarify what the CAMTA3 (A855V) mutant is expected to do in this context (constitutively (in)active?). I don't really see how the

authors can make many conclusions on line 253-255 based on the observation that CAMTA transgenics have normal MPK phosphorylation patterns?

8) The highlight of the study are for me clearly the images/video's showing the Ca²⁺ waves after triggering a trichome, which are very nice to watch. Also the transcriptome comparing Col-0 with glabrous gl1 during touch further brings home the point that trichomes make plants more sensitive to mechanical stimulation, basically acting like antennae or extensions to maximise mechanical impact on the surrounding skirt cells. Nevertheless, the role of trichomes as mechanosensors and initiation of Ca²⁺ spikes in cells surrounding trichomes after touching has been shown previously in Arabidopsis (Zhou et al. 2017, PCE <https://pubmed.ncbi.nlm.nih.gov/26920667/>). Also in Venus flytrap the trigger hairs are known to be trichomes, and were previously shown to induce a calcium wave in cells surrounding the triggered trichome by Suda, Nature Plants, 2020 <https://pubmed.ncbi.nlm.nih.gov/33020606/>

Minor comments

1) Trichomes undoubtedly help plants with amplifying mechanical cues, and the authors present supporting data the gl1 glabrous mutants do not show improved resistance to infection after mechanostimulation (Fig 5f-g). Chehab et al., 2012 previously showed the role of JA synthesis in this kind of touch-induced resistance using the aos mutant, which is interestingly also in the gl1 background. In their study the gl1 mutant was used as a control for touch-induced pathogen/pest resistance compared to the aos (gl1) mutant and showed clear touch induced pathogen resistance effects. These findings should be discussed and compared to the authors' currently presented work.

Reviewer #3 (Remarks to the Author):

Matsumura et al. describe plant immune response triggered by mechanical stimuli through trichome cells in Arabidopsis thaliana. Mechanical stimuli (MS) such as artificial rain or brush treatment were found to induce expression of mechanosensitive genes and plant genes associated with immunity. CAMTA3 was identified as a regulator of MS-mediated gene expression and immune priming. In agreement with stimulating plant immune response by mechanical stimuli, pretreatment of artificial rain drop or mechanical stimulus by brushing suppressed Alternaria brassicicola infection in Arabidopsis Col-0 plants. Similar, mechanical stimulus pretreatment also suppressed biotrophic pathogen Pseudomonas syringae pv. maculicola ES4326 infection. The authors also found that intercellular calcium wave is initiated from trichomes following MS and that trichomes may be involved in primed immune response by MS.

Overall, this manuscript shows an interesting immune priming phenomenon through trichome sensing of MS, which will be of interest to the readership of Nature communication. However, a few new experiments are needed to ascertain the main conclusion.

Major points

1. Detailed information about the *gl1* mutant (e. g. allele) used in this study is missing. We were not able to find any related information from the citations (#36 and #38). More importantly, *gl1*, *gl3*, and *ttg1* mutants not only have a defect in trichome development, but also have defects in systemic acquired response and some aspect of the cuticle structure (Xia et al. 2010, *Plant physiol.*). As such, data presented are not sufficient to conclude that lack of trichome per se is responsible for failed priming by MS in the *gl1* mutant. At a minimum, the authors need to examine a comprehensive set of cuticle mutants as well as defense mutants that are phenotypically associated with *gl1* and other trichome development mutants to determine a direct or indirect effect of trichomes in MS-mediated calcium signal and immune response before making the most important conclusion of this paper.

2. Experiments showing equal expression levels of GCaMP3 protein in transgenic plants (35Spro:GCaMP3 in Col-0 and *gl1*) are necessary.

3. Is syringe-infiltration method appropriate to assess immune priming by MS? Syringe-infiltration involves MS, right(?)

Minor points

Line 156: "Data not shown" is not appropriate. Please include the dataset.

Figure 2f. Dataset confirming genotypes of *fls2* and *bak1-3* mutants is necessary.

Reviewer #1 (Remarks to the Author):

In this manuscript authors report that *Arabidopsis* trichomes can act as mechanosensory structures that can activate a wide range of immune responses. It is a comprehensive study and connects the mechanical input plants experience in nature (raindrops) to the immune resistance, identifying signalling factors (Ca²⁺, MAPKs, hormones, transcription factors, and their target genes) in between. The writing is concise and clear and successfully summarizes (what must have been) years of work succinctly.

I have few points to suggest changes that can improve the manuscript further.

1) This report emphasizes raindrop as a possible trigger of immune responses through its mechanical stimulating effect. While this is plausible, this begs for further experimentations such as: a) analysis of pathogen content of raindrops, and b) comparison of artificial raindrop experiment and MS treatment to plants under natural rain (more pathogen response?). This additional sample is wanted in all analysis in Fig 1, 2, and 4. Alternatively, it can be written from a slightly shifted angle that mechanical stimulation of the trichome (including raindrops) triggers defence response. This shift would also emphasize the broader applicability of the findings; they have uncovered general mechano-induction of defence responses that is trichome-dependent.

Authors' response: We appreciate the reviewer's comments. As to "a) analysis of pathogen content of raindrops", it has been reported that natural raindrops contain bacteria at the concentration of 1.06×10^4 (/cm³) (Casareto et al., 1996, *Geophys. Res. Lett.*), including plant pathogenic bacteria such as *Pseudomonas syringae* (Pratanth et al., 2015, *Int. Res. J. Biol. Sci.*; Lu et al., 2016, *Aerosol and Air Qual. Res.*), *Xanthomonas campestris*, and *Pantoea ananatis* (Schwartz et al., 2003, *Plant Dis.*). Likewise, it has been also found that rainwater contains fungi such as *Alternaria* sp., *Fusarium* sp., *Cladosporium* sp., *Phoma* sp., *Rhizopus* sp., and *Botrytis cinerea* (Palmero et al., 2011, *J. Ind. Microbiol. Biotechnol.*). We have revised the Introduction section to include this information.

We totally agree with the reviewer's comments in (b) that the comparison of plants after treatment with artificial raindrops and MS to plants under natural rain is very important. Thus, we performed RT-qPCR experiments to determine whether natural rain induces the expression of MS-induced genes similarly to artificial rain. As shown in Supplementary Fig. 1e, the MS-induced genes *TCH4* and *WRKY33* were upregulated after treatment with natural rain. In addition, we performed the pathogen test under natural rain. However, because it is quite difficult to monitor where and how often raindrops hit the leaf surface, we could not obtain scientifically

credible data during the revision period. In addition, as field experiments using transgenic plants are not allowed in Japan, we cannot evaluate the trichome-induced calcium waves under natural rain. Therefore, we revised the Abstract, Results, and Discussion sections according to the reviewer's suggestions that mechanical stimulation of the trichome, including raindrops, triggers defence response. We sincerely appreciate the reviewer's comments as the text now can emphasize the general impact of mechano-sensitive immunity in *Arabidopsis thaliana*.

2) Since mechanical stimulation of the trichomes seems to be the critical trigger, it would be helpful to have explicit estimation of how much stress or strain the trichome experiences in receiving a raindrop or the particular types of brushing.

Authors' response: Thank you for the useful comments. As suggested, we have measured the force density (N/m²) applied to the leaf surface by falling droplets and brushing. As shown in new Supplementary Figure 4, the force densities of one falling droplet and one brushing are 22.9 ± 3.1 and 247.7 ± 59.3 N/m², respectively. This is consistent with our findings that 10 falling droplets ($22.9 \text{ N/m}^2 \times 10$) and one brushing (247.7 N/m^2) induce the expression of almost the same set of genes to similar levels, suggesting that these two mechanical stimuli applied similar levels of physical impact to the leaf surface.

3) Note that the mechano stimulation used in this study is quite specific – seemingly selected with the mechano-sensation by trichomes in mind (raindrops and brushing). There are other types of physiologically relevant mechanical stimulation, such as bending of the leaf or stem, stomping of the whole plants, wind effects, and so on, and this should be accounted in some sentences (e.g., L157: it's not just duration or intensity; L302 – main sensor of this type of MS, or say senses mechanical stimulation on the leaf surface).

Authors' response: As suggested by the reviewer, we have revised the sentences as follows:
L86, “Here, we report a novel layer of the plant immune system evoked by sensing mechanostimulation on the leaf surface.”
L174, “which are differentially activated by MS depending on their type, intensity, and duration, and on which organs perceive the stimulation.”
L339, “trichomes serve as sensor of this type of MS on the leaf surface.”

4) How much more susceptible are *gl* mutants to pathogens?

Authors' response: A previous study demonstrated that the *gl1*, *gl3*, and *ttg1* mutants, which do not form trichomes, are unable to mount systemic acquired resistance (SAR) (Xia et al., 2010,

Plant Physiol.), a secondary immune response in uninfected distal tissues after a primary infection in a local site, while basal levels of immunity are not dramatically affected (Fig. 5g, 5h, Supplementary Fig. 11, Xia et al., 2010, Plant Physiol.). The authors found that the defective SAR in these trichome mutants was associated with impairment in their cuticle formation, but not trichomes. Indeed, Xia et al. demonstrated that *cer1-1* and *cer3-1* (*wax2*) mutants, which have defects in synthesizing cuticular wax but form normal trichomes, fail to induce SAR (Xia et al., 2009, Cell Host & Microbe). Therefore, the immunodeficient phenotype of *gl* mutants is due to disordered cuticle formation. To investigate whether trichome-mediated immunity is compromised in *cer1* and *cer3* mutants, we performed the pathogen test using the necrotrophic *Alternaria brassicicola* Ryo-1 after mechanical stimulation of the leaf surface of these mutants (new Supplementary Fig. 12). Surprisingly, pretreatment with brushing significantly suppressed lesion development compared to control plants without pretreatment. These data strongly indicate that the lack of trichome in *gl* mutants is responsible for failed MS-induced immunity.

5) L344: My recollection is that trichomes are not monophyletic in all plant species – so conserved is a strong word. Replace with found?

Authors' response: We sincerely appreciate the reviewer's comments. We revised the sentence as follows: "As the trichome on the leaf surface is widely **found** in many land plants, there may be a common and novel intercellular network of cell–cell communication that initiates calcium waves for activating immune responses".

6) More details of the RNA-seq is needed. How many replicates per sample type? Read depth?

Authors' response: According to the reviewer's comment, we now added details of RNA-seq analysis. Each experiment had three biological replicates per sample with approximately 10 million 81-nt reads. We added this information to the sentences in L103, L127 and L335: "(three biological replicates, \log_2 fold changes (\log_2FC) ≥ 1 , $P < 0.05$)".

Reviewer #2 (Remarks to the Author):

In this work by Matsumara et al., a wide range of experiments and meta-analyses are presented that add further insight into how plants respond to mechanical stimulation. The title focuses on the role of trichome cells in immune responses, but this topic is only addressed in the last two results sections, and various other data is presented earlier on in the manuscript. Though in principle all the findings are of interest, in my opinion many of the findings are not conceptual

advances and are rather confirmatory or adding incremental advances. Moreover, the manuscript is written in a style that would suggest the uninformed reader that many findings are novel, while they are to a large extent conceptually part of the published literature. The manuscript is clearly written and the figures are well presented.

Major comments.

1) The manuscript presents a lot of findings that are confirmatory or supplemental to existing studies. For instance, the concept that the impact of water drops on leaves causes an overlapping response to 'classical' mechanical touching e.g. with a brush is well-known already for decades (e.g. Braam et al., Cell, 1990), and confirmed by many studies.

Authors' response: We sincerely appreciate the reviewer's comments. Although we cited the indicated previous work, we realized that more information on water drop- and touch-induced responses was needed to clarify the novelty of our study. We revised the main text by describing relevant works.

The dose response analysis of number of drops to gene expression shows one drop is largely sufficient and two drops results in very similar effects as 10 drops. The first section of the results describes an RNA seq analysis of the response after 15 minutes to impact of 10 drops of water from about 120 cm high onto the leaf. This is in principle a comparable experiment to the RNA-seq analysis performed previously with water spray bottles at various time points from 10-120 minutes (Van Moerkercke et al., 2019, PNAS). Brushing RNA-seqs have also been performed and published recently e.g. Xu et al., Plant Journal, 2019 (<https://pubmed.ncbi.nlm.nih.gov/30537160/>) and Wang et al., PNAS, 2018 (<https://www.ncbi.nlm.nih.gov/pmc/articles/PMC6205429/>). The comparative analysis of the authors own-data with these published data sets would be useful here. I could also not find details regarding fold-change/p-value cut-offs and multiple testing corrections for the authors new data set in the results section or methods.

Authors' response: According to the reviewer's suggestions, we have performed the comparative analysis of our own data with published datasets as shown below (new Supplementary Fig. 3). We found that the percentages of overlapping genes expressed in both previous and our studies are 43.5% (a; Van Moerkercke et al. water spray/our raindrop), 62.4% (b; Van Moerkercke et al. brushing/our brushing), 29.7% (c; Lee et al. bending/our brushing), and 40.7% (Xu et al. brushing/our brushing), while our raindrop- and brushing-induced genes overlapped 87.3% (our brushing/our raindrop). These differences could be largely due to 1) type

of mechanostimulation (e.g., bending, brushing, or wounding), 2) intensity and time of stimulation, 3) plant growth conditions (e.g., MS- or soil-grown), 4) type of tissue to which stimulation was applied.

We have measured the force density (N/m^2) applied to the leaf surface by falling droplets and brushing. As shown in new Supplementary Fig. 4, the force densities of one falling droplet and one brushing are 22.9 ± 3.1 and $247.7 \pm 59.3 \text{ N/m}^2$, respectively. This is consistent with our findings that 10 falling droplets ($22.9 \text{ N/m}^2 \times 10$) and one brushing (247.7 N/m^2) induce the expression of almost the same set of genes to similar levels, suggesting that these two mechanical stimuli applied similar levels of physical impact to the leaf surface. To our knowledge, this is the first study that quantitatively adjusts the force to apply artificial raindrops to the leaf surface.

2) The meta-analysis comparing the water drop/MS transcriptomes with published data sets focusing on e.g. flg22 pseudomonas and SA/JA is interesting, but again not very novel, as similar analyses have been done.

Authors' response: As the reviewer mentioned, the Gene Ontology analysis using brushing-induced dataset has been done by Xu et al. (Plant Journal, 2019), demonstrating that JA, GA, and calcium signals have been implicated in brushing-induced mechanotransduction. On the other hand, dynamic transcript profiling of the water spray-induced genes has been shown by Van Moerkercke et al. (PNAS, 2019). The authors confirmed that JA signaling plays a key role in responses to water spray. Our meta-analysis strongly indicated that raindrop-induced genes are regulated mainly by immune signaling pathways (Fig. 2a and Supplementary Fig. 2). However, as shown in new Supplementary Fig. 6, we found that JA-responsive MYC transcription activators are not major regulators in early gene expression induced by brushing and raindrops under our experimental conditions. To support these results, we further demonstrated that a JA-insensitive mutant, *jai3-1*, that produces a truncated form of JAZ3 repressor protein also expressed the mechanosensitive genes such as *TCHs*, *WRKYs*, *CBP60g*, *MYB51*, *JAZ1*, and *ERF1*, concluding a novel aspect of mechanotransduction.

Fig. 1

Supplementary Fig. 3

Furthermore the methods section lacks a lot of detail on how these experiments were performed on different platforms were analysed (e.g. software packages, parameters for fold-changes/p-values). A table giving the precise experimental conditions would be needed, as many treatments like spraying with chemicals like SA/Flg22 would trigger mechanoresponses in themselves. Percentage overlaps between data sets are presented, but there is no statistical analysis whether

such overlaps are statistically speaking meaningful or possible by random distributions.

Authors' response: We apologize for the inconvenience. We have now provided the information requested by the reviewer in new Supplementary Table 11. As to percentage overlaps between datasets in Fig. 2a, the individual dataset has been statistically processed as shown in new Supplementary Table 11, and the overlaps are meaningful.

3) The hormone analysis again largely confirms the findings of Van Moerkercke et al., 2019 that mainly JA and JA-Ile are early touch responsive. I could however not find indications of statistical significance in the graphs of Fig 1C. The very early 5 min time point is interesting, but further time points should have been taken to match the RNAseq data. Also a strong drop in ABA concentration was measured 10-25 minutes after water spray previously by Van Moerkercke et al., which likely is caused by the water excess due to water spray/drops. This effect was not found here after 5 minutes, but may become apparent some minutes later. As stated by the authors in the introduction, this may affect immune responses e.g. via stomatal opening allowing pathogens to enter, and would be thus important to confirm using the counted water drop system. The GA data should be shown (even if supplementary).

Authors' response: As suggested by the reviewer, we have newly performed the mass spectrometry measurement of phytohormones stimulated 15 min after applying brush, raindrops, and wounding to 4-week-old plants. Similar to Fig. 2c, JA and JA-Ile were slightly but significantly accumulated in response to raindrop and brushing. However, the other hormones, including ABA, neither increased nor decreased. As to the ABA reduction observed by Van Moerkercke et al., we think that our experimental condition is quite different from that of Van Moerkercke et al. The authors kept MS-grown plants under a high humidity condition after water spray, whereas we used soil-grown plants and kept them at 50–70% humidity as described in the Methods section. The data on GA₄ is shown in new Supplementary Fig. 5.

4) I strongly disagree with the statement on line 152-154, saying that most mechanosensitive genes, whose expression is induced after 5 minutes, are presumably regulated independently of phyto hormones. First of all, it is unclear which specific genes induced after 5 minutes are referred to here, as the authors' RNA-seq data set is done on 15 minutes and the selected WRKY33 reporter gene is also not yet consistently induced by rain drops after 5 minutes (it is induced after 10 min) (Supp figure 1d). Furthermore, JA/JA-Ile are clearly induced by water drops/MS at the hormone level even after 5 min (though stats are lacking) in Figure 1c.

Authors' response: We appreciate the important comment made by the reviewer. We have shown new phytohormone measurements obtained 15 min after treatment of WT with raindrops, brushing, and wounding (new Supplementary Fig. 5). These data show that only JA and JA-Ile are significantly increased. However, as shown in new Supplementary Fig. 6, the MYC activators are not involved in the expression of *TCHs*, *WRKYs*, *CBP60g*, *MYB51*, *JAZ1*, and *ERF1*, which are thought to be mechanosensitive genes. We agree that JA and JA-Ile could be involved in a part of mechanosensitive gene expression, and we toned down our conclusion in the main text.

5) The data showing that MPK3/6 are phosphorylated by raindrops/MS within minutes is interesting, but was again already shown for MPK4/6 one minute after MS by Ishimura et al., 2008 (<https://onlinelibrary.wiley.com/doi/full/10.1046/j.1365-313x.2000.00913.x>). The finding that phosphorylation does not seem to operate via FLS2 and BAK1 receptors is of interest. Also the suggestion on line 182 that MAPKs play a critical role in mechanotransduction is very circumstantial based on the presented data, as well as previously suggested in many other studies. Wang et al., 2018, PNAS also showed the importance of phosphorylation of e.g. TREP1 for mechanosignalling.

Authors' response: We appreciate the recommendation of these important papers. We have now described these studies in the text. Notably, to my knowledge, we have shown for the first time that the calcium ionophore A23187 rapidly stimulates MAPK phosphorylation, linking mechanostimulation to the kinase cascade (Fig. 2d, 2e and Supplementary Fig. 8g).

6) Mechanostimulation has extensively been shown to trigger resistance to necrotrophic pathogens and insect pests (e.g. Chehab et al., 2012, Current Biology; Choi et al. 2017, Sci Rep; summarised by Ghosh et al., 2021, J Exp Bot). The addition of a biotrophic pathogen like *Pseudomonas* to this list is of course interesting, but again confirmatory. The similarity with chitin-induced transcriptomic responses was also reported already by Xu et al., 2019, Plant J.

Authors' response: We thank the reviewer for the information. We have now further described the finding from the Chehab et al. paper that bending treatment enhances resistance to *Botrytis cinerea*. It has been reported that natural raindrops contain bacteria at the concentration of 1.06×10^4 (/cm³) (Casareto et al., 1996, Geophys. Res. Lett.), including plant pathogenic bacteria such as *Pseudomonas syringae* (Pratanth et al., 2015, Int. Res. J. Biol. Sci.; Lu et al., 2016, Aerosol and Air Qual. Res.), *Xanthomonas campestris*, and *Pantoea ananatis* (Schwartz et al., 2003, Plant Dis.). Likewise, it has been also found that rainwater contains fungi such as *Alternaria* sp., *Fusarium* sp., *Cladosporium* sp., *Phoma* sp., *Rhizopus* sp., and *Botrytis cinerea* (Palmero et al.,

2011, J. Ind. Microbiol. Biotechnol.). Therefore, as raindrops deliver both biotrophs and necrotrophs, it would be counterintuitive for mechanostimulation to induce only the JA-mediated defence response, which confers resistance to necrotrophs but promotes biotroph infection. Now, our work has shown that mechanostimulation applied by raindrops activates resistance to diverse pathogens with different parasitism.

7) The finding that CAMTA3 transcription factor is involved in mechanosensing is probably among the most interesting finding of the study. The authors perform meta-analysis of previously reported microarray data and find an overlap between *camta123* regulated genes and their own raindrop/MS transcriptomes. An overrepresentation of the *camta* DNA binding site in touch induced promoters (Fig 3a) was already reported by Xu et al., 2019, Plant J. The Chip-seq on CAMTA3 is of clear interest and forms one of the most interesting data sets of this paper.

Authors' response: According to the reviewer's comment, we have cited the Xu et al. paper and revised the main text as follows: "From an unbiased promoter analysis of the top 300 genes among 917 differentially expressed genes, we obtained the highest enrichment for the CGCG box (CGCGT or CGTGT), which is recognized by calmodulin (CaM)-binding transcription activators (CAMTAs) that are conserved from plants to mammals (Fig. 3a). A similar motif analysis that detects the CGCG box among the brushing-induced gene promoters was reported by Xu et al."

The description of the different mutants/complementations of CAMTA transgenics could be improved to clarify what the CAMTA3 (A855V) mutant is expected to do in this context (constitutively (in)active?). I don't really see how the authors can make many conclusions on line 253-255 based on the observation that CAMTA transgenics have normal MPK phosphorylation patterns?

Authors' response: According to the reviewer's comment, we have revised the main text as follows: "The CAMTA3^{A855V} transgenic plants, which possesses a mutation in the IQ domain, have been reported to suppress the constitutive expression of defence-related genes seen in the *camta2 camta3* double mutant, and are no longer regulated by calcium-mediated responses".

I don't really see how the authors can make many conclusions on line 253-255 based on the observation that CAMTA transgenics have normal MPK phosphorylation patterns?

Authors' response: According to the reviewer's comment, we have revised the main text as follows: "These results suggested that the mechanotransduction initiated by raindrops and MS may cause a Ca²⁺ influx that negates the repressive effect of CAMTA3 and independently activates the MAPK cascade, as previously proposed".

8) The highlight of the study are for me clearly the images/video's showing the Ca²⁺ waves after triggering a trichome, which are very nice to watch. Also the transcriptome comparing Col-0 with glabrous gl1 during touch further brings home the point that trichomes make plants more sensitive to mechanical stimulation, basically acting like antennae or extensions to maximise mechanical impact on the surrounding skirt cells. Nevertheless, the role of trichomes as mechanosensors and initiation of Ca²⁺ spikes in cells surrounding trichomes after touching has been shown previously in Arabidopsis (Zhou et al. 2017, PCE <https://pubmed.ncbi.nlm.nih.gov/26920667/>).

Authors' response: We greatly appreciate the reviewer's comments. The Zhou et al. paper demonstrated that, compared to our data, touching trichomes induces the calcium influx only in the surrounding skirt cells but not further away from them. This is quite different from our finding as our study unambiguously demonstrated that the expanding Ca²⁺ waves enable plants to activate immune response in the very same regions (this is illustrated in Fig. 6).

Also in Venus flytrap the trigger hairs are known to be trichomes, and were previously shown to induce a calcium wave in cells surrounding the triggered trichome by Suda, Nature Plants, 2020 <https://pubmed.ncbi.nlm.nih.gov/33020606/>

Authors' response: We appreciate the reviewer's comments. We have mentioned this finding from the Venus flytrap in the last paragraph.

Minor comments

1) Trichomes undoubtedly help plants with amplifying mechanical cues, and the authors present supporting data the gl1 glabrous mutants do not show improved resistance to infection after mechanostimulation (Fig 5f-g). Chehab et al., 2012 previously showed the role of JA synthesis in this kind of touch-induced resistance using the aos mutant, which is interestingly also in the gl1 background. In their study the gl1 mutant was used as a control for touch-induced pathogen/pest resistance compared to the aos (gl1) mutant and showed clear touch induced pathogen resistance effects. These findings should be discussed and compared to the authors' currently presented work.

Authors' response: Chehab et al. mechanically stimulated leaves by bending them back and forth 10 times, which induced JA-dependent resistance to *Botrytis cinerea*. Because bending treatment directly applies the mechanical stimuli inside the leaf including mesophyll cells, it is plausible that the *gll* mutant induces resistance to the pathogen in response to mechanical stimuli. We have now discussed this in the main text L346: “Mechanically stimulating leaves by bending them back and forth 10 times induces JA-dependent resistance to *B. cinerea* in both Col-0 and *gll* mutant, suggesting that bending directly stimulates cells inside the leaf.”

Reviewer #3 (Remarks to the Author):

Matsumura et al. describe plant immune response triggered by mechanical stimuli through trichome cells in *Arabidopsis thaliana*. Mechanical stimuli (MS) such as artificial rain or brush treatment were found to induce expression of mechanosensitive genes and plant genes associated with immunity. CAMTA3 was identified as a regulator of MS-mediated gene expression and immune priming. In agreement with stimulating plant immune response by mechanical stimuli, pretreatment of artificial rain drop or mechanical stimulus by brushing suppressed *Alternaria brassicicola* infection in *Arabidopsis Col-0* plants. Similar, mechanical stimulus pretreatment also suppressed biotrophic pathogen *Pseudomonas syringae* pv. *maculicola* ES4326 infection. The authors also found that intercellular calcium wave is initiated from trichomes following MS and that trichomes may be involved in primed immune response by MS.

Overall, this manuscript shows an interesting immune priming phenomenon through trichome sensing of MS, which will be of interest to the readership of *Nature communication*. However, a few new experiments are needed to ascertain the main conclusion.

Major points

1. Detailed information about the *gll* mutant (e. g. allele) used in this study is missing. We were not able to find any related information from the citations (#36 and #38).

Authors' response: We apologize for this oversight. We now added information about the *gll* mutant used in this study. We used Col(*gll*) obtained from Lehle Seeds (TX, USA), which was previously designated as *gl-1* mutant (Koornneef et al., 1982, *Thor. Appl. Genet.*). It has been reported that *gl-1/Col(gll)* is hairless and possesses the mutation in chromosome 3 (Koornneef et al., 1982, *Thor. Appl. Genet.*).

More importantly, *gl1*, *gl3*, and *ttg1* mutants not only have a defect in trichome development, but also have defects in systemic acquired response and some aspect of the cuticle structure (Xia et al. 2010, Plant Physiol.). As such, data presented are not sufficient to conclude that lack of trichome per se is responsible for failed priming by MS in the *gl1* mutant. At a minimum, the authors need to examine a comprehensive set of cuticle mutants as well as defense mutants that are phenotypically associated with *gl1* and other trichome development mutants to determine a direct or indirect effect of trichomes in MS-mediated calcium signal and immune response before making the most important conclusion of this paper.

Authors' response: This is an excellent point. A previous study demonstrated that the *gl1*, *gl3*, and *ttg1* mutants, which do not form trichomes, are unable to mount systemic acquired resistance (SAR) (Xia et al., 2010, Plant Physiol.), a secondary immune response in uninfected distal tissues after a primary infection in a local site, while basal levels of immunity are not dramatically affected (Fig. 5, Supplementary Fig. 11, Xia et al., 2010, Plant Physiol.). Importantly, the authors found that the defective SAR in these trichome mutants is associated with impairment in their cuticle formation, but not trichomes. Indeed, Xia et al. demonstrated that *cer1-1* and *cer3-1* (*wax2*) mutants, which have defects in synthesizing cuticular wax but form normal trichomes, fail to induce SAR (Xia et al., 2009, Cell Host & Microbe). Therefore, the immunodeficient phenotype of *gl* mutants is due to disordered cuticle formation. To investigate whether trichome-mediated immunity is compromised in *cer1* and *cer3* mutants as in *gl1*, we performed the pathogen test using the necrotrophic *Alternaria brassicicola* Ryo-1 after mechanical stimulation of the leaf surface of these mutants (new Supplementary Fig. 12). Surprisingly, pretreatment with brushing significantly suppressed lesion development compared to the control plants without pretreatment. These data strongly indicate that the lack of trichome, but not cuticle, in *gl* mutants is responsible for failed MS-induced immunity.

In addition, we investigated the requirement for MS-induced immunity in the defence mutants *fls2*, *bak1*, *npr1*, and *myc2/myc3/myc4*. As shown in new Supplementary Fig. 16, FLS2 and NPR1 could be involved in the signaling pathway; however, JA-responsive MYC activators are unlikely to be direct mediators.

We revised the main text to include these results.

2. Experiments showing equal expression levels of GCaMP3 protein in transgenic plants (35Spro:GCaMP3 in Col-0 and *gl1*) are necessary.

Authors' response: We appreciate the reviewer's comments. We confirmed the equal accumulation of GCaMP3 protein in both *35Spro:GCaMP3/Col-0* and *35Spro:GCaMP3/gl1*

transgenic plants (new Supplementary Figure 13).

3. Is syringe-infiltration method appropriate to assess immune priming by MS? Syringe-infiltration involves MS, right(?)

Authors' response: As the reviewer commented, syringe infiltration involves MS. However, as shown in the inoculated leaf, when we pressure-infiltrate *Psm* ES4326 in a whole leaf with a syringe, mechanical stress is only applied in the indicated infection site (red circle). Because the infected leaf disk does not overlap with the infection site, and MS activates immunity only in the contact sites (Fig. 4a), the effect of syringe infiltration on the induction of immunity is likely to be minimum. In addition, mock control (without MS pretreatment) also is pressure-infiltrated with the pathogen, we think that the effect of raindrops and brushing can be appropriately evaluated.

Minor points

Line 156: “Data not shown” is not appropriate. Please include the dataset.

Authors' response: We apologize for the inconvenience. We now added the dataset of gibberellin measurement obtained upon application of raindrops or MS (new Supplementary Fig. 5). As demonstrated in the figure, GA₄ was not detected in the treated samples.

Figure 2f. Dataset confirming genotypes of *fls2* and *bak1-3* mutants is necessary.

Authors' response: As the reviewer suggested, we included the dataset confirming genotypes of *fls2* (SALK_093905) and *bak1-3* (SALK_034523) (new Supplementary Fig. 7).

REVIEWERS' COMMENTS

Reviewer #1 (Remarks to the Author):

I found the reviewers' comments well considered and reflected in the revised manuscript.

It is nice to see the force density quantified now; however, I don't find the protocol on how this has been done. Please add this in the Materials and Method.

Reviewer #2 (Remarks to the Author):

The reviewers have thoroughly revised the manuscript and addressed most of my (Reviewer 2) comments satisfactorily.

There are however a few relatively small points that were not addressed fully or correctly, which should be addressed in the final version.

In response to Comment 1)

I am happy to see that the authors included a meta-analysis with the suggested other MS transcriptome data sets. However, I do not really understand what the authors mean by the statement that they do 'not observe strong correlations with bending and brushing treatments'. Judging from the new Venn diagrams in Suppl. Figure 3, 385 genes of 760 'bending' induced genes reported in Lee et al., 2005, are also induced by the mechanostimulation (MS) in the author's own work, accounting for >50%. Also 513 out of 1007 brushing-induced genes reported by Xu et al., 2019, again >50%, are common with the new MS data presented here. To me these sound like strong overlaps, which are undoubtedly statistically significant. Also if the term correlation is used, especially given that the authors provide Pearson correlations in the line before comparing their own raindrop and MS datasets, the authors should report the Pearson coefficient for these new comparisons to support their statements. I don't think it reduces the validity of their own work in any way that there are clear (and expected) overlaps with previous MS-related transcriptomes, so it is absolutely fine if there are overlaps/correlations. However, the reporting should be clear and consistent. Also it should be cited in the text which data sets were used for this meta-analysis, and referred to Suppl Table 11 for the precise info on the data sets.

As a small note, I believe that the Van Moerkercke 2019 data used for Fig S3B is also based on water-spray, and not on brushing as shown in the figure. This should be adjusted in Fig S3B and Supp Table 11.

In response to Comment 2)

I am thankful to the authors to add the Suppl Table 11 with the experimental conditions used for the meta-analysis. The authors say in the rebuttal that a statistical analysis is provided in Suppl Table 11 to support overlaps between data sets in Fig 2a, stating that the overlaps are 'meaningful. Though I trust the author's explanation, I could not find any statistical information in Suppl Table 11. This should be corrected. Also, I do not see any mention of this statistical analysis in the results section line 144-153. This should thus be addressed appropriately.

In response to Comment 3)

I could not find any reference to the suggested Nishimura 2008 paper in the context of MPK4/6 phosphorylation after MS, although the authors mentioned that they did in the rebuttal. The authors seem to cite Wang et al., 2018 PNAS here, which I believe should be the Nishimura 2008 paper. This should be corrected.

Reviewer #3 (Remarks to the Author):

The authors have addressed my major concerns.

Response to Reviewer's comments

Reviewer #1 (Remarks to the Author):

I found the reviewers' comments well considered and reflected in the revised manuscript.

Authors' response: We sincerely express our appreciation to Reviewer #1 for your suggestions and comments.

1) It is nice to see the force density quantified now; however, I don't find the protocol on how this has been done. Please add this in the Materials and Method.

Authors' response: We apologize for the inconvenience. We described the protocol for the quantification of the force density in the Materials and Method: "The abaxial side of leaves from 4-week-old plants was physically attached to the measuring pan of the electronic balance QUINTIX224-1S (Sartorius Lab Instruments GmbH & Co., Göttingen, Germany) with surgical tape (3M Company, MN, USA). The adaxial side of the leaf was treated with 1 falling droplet or brushed once (shown in the above subsections). The peak weight applied to the leaf surface was obtained as the force. The force per unit area (N/m^2) is converted from the peak weight (kg) and the contact area of the brush tip ($5.6 \times 10^{-5} \text{ m}^2$) or raindrop ($9.73 \times 10^{-6} \text{ m}^2$)."

Reviewer #2 (Remarks to the Author):

The reviewers have thoroughly revised the manuscript and addressed most of my (Reviewer 2) comments satisfactorily.

Authors' response: We thank Reviewer #2 for helpful comments to improve our study.

There are however a few relatively small points that were not addressed fully or correctly, which should be addressed in the final version.

In response to Comment 1)

1) I am happy to see that the authors included a meta-analysis with the suggested other MS transcriptome data sets. However, I do not really understand what the authors mean by the statement that they do ‘not observe strong correlations with bending and brushing treatments’. Judging from the new Venn diagrams in Suppl. Figure 3, 385 genes of 760 'bending' induced genes reported in Lee et al., 2005, are also induced by the mechanostimulation (MS) in the author's own work, accounting for >50%. Also 513 out of 1007 brushing-induced genes reported by Xu et al., 2019, again >50%, are common with the new MS data presented here. To me these sound like strong overlaps, which are undoubtedly statistically significant. Also if the term correlation is used, especially given that the authors provide Pearson correlations in the line before comparing their own raindrop and MS datasets, the authors should report the Pearson coefficient for these new comparisons to support their statements. I don't think it reduces the validity of their own work in any way that there are clear (and expected) overlaps with previous MS-related transcriptomes, so it is absolutely fine if there are overlaps/correlations. However, the reporting should be clear and consistent. Also it should be cited in the text which data sets were used for this meta-analysis, and referred to Suppl Table 11 for the precise info on the data sets.

Authors' response: We totally agree to provide the Pearson coefficient for new comparisons of transcriptome data and now have shown in the main text as follows: “In addition, we observed clear correlations between our profiles and previously published datasets obtained by water spray, bending, brushing, and cotton swabbing treatments (Supplementary Fig. 3 and Supplementary Data 11).”

2) As a small note, I believe that the Van Moerkercke 2019 data used for Fig S3B is also based on water-spray, and not on brushing as shown in the figure. This should be adjusted in Fig S3B and Supp Table 11.

Authors' response: We apologize for the inconvenience that we made a mistake in the description of the Van Moerkercke 2019 data. We revised the figure and explanation in Supplementary Fig. 3b and Supplementary Data 11.

In response to Comment 2)

I am thankful to the authors to add the Suppl Table 11 with the experimental conditions used for the meta-

analysis. The authors say in the rebuttal that a statistical analysis is provided in Suppl Table 11 to support overlaps between data sets in Fig 2a, stating that the overlaps are ‘meaningful. Though I trust the author’s explanation, I could not find any statistical information in Suppl Table 11. This should be corrected. Also, I do not see any mention of this statistical analysis in the results section line 144-153. This should thus be addressed appropriately.

Authors’ response: As pointed by the reviewer, we made a mistake in the description of our previous rebuttal letter. To show the Venn diagrams in Fig. 2a, a statistical analysis cannot be performed as we did for Fig. 1f. Therefore, we did not make any changes in the main text for the explanation of Fig. 2a.

In response to Comment 3)

I could not find any reference to the suggested Nishimura 2008 paper in the context of MPK4/6 phosphorylation after MS, although the authors mentioned that they did in the rebuttal. The authors seem to cite Wang et al., 2018 PNAS here, which I believe should be the Nishimura 2008 paper. This should be corrected.

Authors’ response: The reviewer's comment is correct. We apologize for the inconvenience. We now cited the reference, Ichimura et al., 2000, Plant J., which was initially suggested by the reviewer in the first review as follows: “The data showing that MPK3/6 are phosphorylated by raindrops/MS within minutes is interesting, but was again already shown for MPK4/6 one minute after MS by Ishimura et al., 2008 (<https://onlinelibrary.wiley.com/doi/full/10.1046/j.1365-313x.2000.00913.x>)”. The hyperlink shown by the reviewer is for the Ichimura et al., 2000, Plant J. paper (but not for Ishimura et al., 2008), and we found that the suggested manuscript is a very important citation for our work.

Reviewer #3 (Remarks to the Author):

The authors have addressed my major concerns.

Authors’ response: We sincerely thank Reviewer #3 for critical and constructive comments.